# Fast-Slow Thinking GRPO for Large Vision-Language Model Reasoning

**Wenyi Xiao**
Zhejiang University
wenyixiao@zju.edu.cn

**Leilei Gan**[†]
Zhejiang University
leileigan@zju.edu.cn

## Abstract

When applying reinforcement learning—typically through GRPO—to large vision-language model reasoning struggles to effectively scale reasoning length or generates verbose outputs across all tasks with only marginal gains in accuracy. To address this issue, we present **FAST-GRPO**, a variant of GRPO that dynamically adapts reasoning depth based on question characteristics. Through empirical analysis, we establish the feasibility of fast-slow thinking in LVLMs by investigating how response length and data distribution affect performance. Inspired by these observations, we introduce two complementary metrics to estimate the difficulty of the questions, guiding the model to determine when fast or slow thinking is more appropriate. Next, we incorporate adaptive length-based rewards and difficulty-aware KL divergence into the GRPO algorithm. Experiments across seven reasoning benchmarks demonstrate that FAST achieves state-of-the-art accuracy with over 10% relative improvement compared to the base model, while reducing token usage by 32.7-67.3% compared to previous slow-thinking approaches, effectively balancing reasoning length and accuracy.

## 1 Introduction

Slow-thinking reasoning has demonstrated remarkable capabilities in solving complex tasks in Large Language Models (LLMs) [1–4] by applying large-scale reinforcement learning (RL), exemplified by OpenAI's o1 [5], DeepSeek-R1 [6], and Qwen's QwQ [7].

Unlike fast-thinking models [8, 9], slow-thinking models undertake more deliberate and thorough reasoning before reaching an answer, which facilitates the exploration of diverse solution paths for a given problem.

Researchers [10–14] have begun exploring similar slow thinking approaches for large vision-language models (LVLMs) to enhance visual reasoning, which can be categorized into SFT-RL two-stage methods [10, 15–18] and RL-only methods [19–22]. SFT-RL methods collect large-scale distilled data from slow-thinking models before applying reinforcement learning, while RL-only methods directly employ reinforcement learning on curated high-quality data.

Despite these efforts, several challenges persist in slow-thinking for LVLM reasoning. First,

Figure 1: FAST achieves higher average accuracy with shorter average response lengths across seven benchmarks. All methods are built upon Qwen2.5-VL.

⌂ https://github.com/Mr-Loevan/FAST
[†]Correspondence to Leilei Gan.

39th Conference on Neural Information Processing Systems (NeurIPS 2025).

while RL-only methods enable slow-thinking LVLMs to improve reasoning accuracy, they struggle to effectively scale reasoning length [19, 21, 17], with observed changes ranging only from –20% to +10% compared to base models. This limited adaptability in reasoning length may constrain their effectiveness on complex tasks. Second, in contrast, we observe that slow-thinking LVLMs with SFT-RL methods [23, 15, 10, 16] exhibit a pronounced overthinking phenomenon—producing overly verbose responses across tasks while yielding only marginal improvements in accuracy. This observation suggests that excessive verbosity may arise from the SFT stage, which performs behavior cloning from distilled data. As evidenced in Table 1, R1-OneVision (one slow thinking model with SFT-RL) produces reasoning chains approximately 2× longer than its base model across all difficulty levels on the Geometry [24] test set. Notably, this overthinking proves detrimental for simpler questions, where extended reasoning results in accuracy degradation (69.5% vs. 72.7%), highlighting the need for adaptive fast-slow thinking.

We notice that current research on addressing the overthinking phenomenon primarily focuses on large language models (LLMs) and can be classified into two categories based on the stage of application. In the training stage, they design length reward shaping in RL training to explicitly encourage concise model responses. [25–28] In the inference stage, they enforce concise reasoning via prompts, e.g., *use less than 50 tokens*, to constrain response length [29, 30]. However, these methods to address overthinking in LLMs ignore challenges of visual inputs and question characteristics in visual reasoning [25–27], leaving their effectiveness in LVLMs largely unexplored. To our knowledge, no existing work effectively balances fast and slow thinking in LVLMs.

Table 1: Comparison of accuracy and response length on Geometry 3K [24] test set across difficulty levels for Qwen2.5-VL-7B, R1-OneVision, and FAST.

| Test | Qwen2.5-VL | | R1-OneVision | | FAST | |
|------|------|------|------|------|------|------|
| | Acc. | Len. | Acc. | Len. | Acc. | Len. |
| Easy | 72.7 | 318 | 69.5 | 623 | 78.2 | 189 |
| Med | 33.9 | 406 | 40.4 | 661 | 49.2 | 220 |
| Hard | 5.5 | 412 | 10.2 | 835 | 12.3 | 304 |
| All | 37.7 | 378 | 40.3 | 731 | 46.4 | 239 |

To address these issues, we propose FAST-GRPO, a tailored variant of GRPO [6, 8] that balances fast and slow reasoning by incorporating adaptive length-based rewards and dynamic regularization conditioned on the characteristics of multimodal inputs. Our approach begins with an investigation of the relationship between reasoning length and accuracy in LVLMs, empirically demonstrating how length rewards and data distributions impact reasoning performance. Based on these findings, our methodology first introduces two complementary metrics to estimate the difficulty of the questions, guiding the model to determine when fast or slow thinking is more appropriate. Next, we incorporate adaptive length-based rewards and difficulty-aware KL divergence into the GRPO algorithm. The former dynamically incentivizes concise or detailed reasoning based on question characteristics, while the latter modulates exploration constraints based on the estimated difficulty of each question.

We conduct extensive experiments on a range of reasoning benchmarks for LVLMs, and the experimental results have demonstrated the effect of the proposed method. As shown in Figure 1, compared with slow thinking or fast thinking methods, our model achieves state-of-the-art reasoning accuracy with an average accuracy improvement of over 10% compared to the base model, while significantly reducing reasoning length against slow thinking models from 32.7% to 67.3%.

## 2 Background: Group Relative Policy Optimization

Group Relative Policy Optimization (GRPO;[6, 8]) extends PPO [31] by replacing the value model with group relative rewards estimation, optimizing the objective in Equation 1.

$$
\mathcal{J}_{GRPO}(\pi_\theta) = \mathbb{E}_{q \sim P(Q), \{o_i\}_{i=1}^G \sim \pi_{\theta_{old}}(\cdot|q)} \left[ \frac{1}{G} \sum_{i=1}^G \frac{1}{|o_i|} \sum_{t=1}^{|o_i|} \left\{ \min \left[ \frac{\pi_\theta(o_{i,t}|q, o_{i,<t})}{\pi_{\theta_{old}}(o_{i,t}|q, o_{i,<t})} \hat{A}_{i,t}, \right. \right. \right.
$$
$$
\left. \left. \left. \text{clip}\left( \frac{\pi_\theta(o_{i,t}|q, o_{i,<t})}{\pi_{\theta_{old}}(o_{i,t}|q, o_{i,<t})}, 1-\epsilon, 1+\epsilon \right) \hat{A}_{i,t} \right] \right\} - \beta D_{KL}\left( \pi_\theta \| \pi_{\text{ref}} \right) \right]
$$
(1)

where $\varepsilon$ and $\beta$ are the clipping hyperparameter and the coefficient controlling the KL regularization [6]. $\hat{A}_{i,t}$ is the advantage, estimated through group relative rewards $\hat{A}_{i,t} = \frac{r_i - \text{mean}(\{r_1, r_2, ..., r_G\})}{\text{std}(\{r_1, r_2, ..., r_G\})}$ with two rule-based rewards: (1) *accuracy reward* ($r_a$) gives a reward when the response is equivalent to the answer, and (2) *format reward* ($r_f$) ensures responses adhere to the specified format.

# 3 Pilot Experiments

As discussed in §1, when applying reinforcement learning—typically through GRPO—to LVLM reasoning struggles to effectively scale reasoning length (RL-only methods; [19, 21, 17]) or generates verbose outputs across all tasks with only marginal gains in accuracy (SFT-RL methods; [23, 15, 10, 16]).

To better understand the factors affecting response length and overall performance in GRPO [6] for LVLM reasoning, we conduct a series of experiments on the Geometry 3K dataset [24]. In particular, we analyze the impact of length-based reward strategies (§3.1) and the influence of data distribution characteristics (§3.2).

## 3.1 Length Rewards Analysis

Prior research has established that while GRPO effectively scales response length in text-only LLMs [6, 20, 32], this effect does not transfer to LVLMs [19, 22]. To verify this phenomenon and explore potential solutions, we performed GRPO-Zero on Qwen2.5-VL [33] with rule-based accuracy reward, and tested ex-

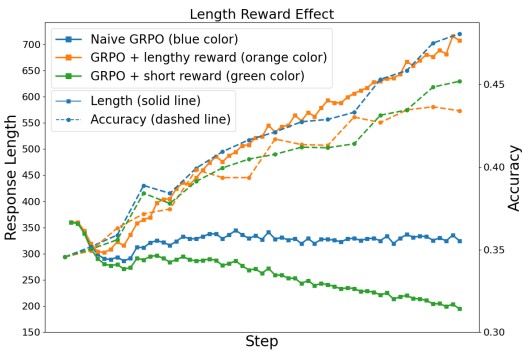

Figure 2: Effect of length rewards on reasoning length and accuracy.

plicit length rewards that either encourage longer correct responses ($r_{\text{lengthy reward}} = L_{correct}/L_{max}$) or shorter correct ones ($r_{\text{short reward}} = 1 - L_{correct}/L_{max}$) as extended rewards, where $L_{max}$ is the maximum token length, $L_{correct}$ denotes the length of the correct response. As shown in Figure 2, with increasing training steps, GRPO with lengthy reward steadily increases to 700 tokens, GRPO with short reward decreases to 180 tokens, while Naive GRPO remains stable around 330 tokens. These length rewards successfully manipulated response length, producing variations from 180 to 700 tokens, but with only modest changes in accuracy (±3%). This decoupling between length and accuracy suggests that LVLMs can maintain reasoning performance across different response lengths, challenging the assumption that longer reasoning is always better.

Based on the findings above, we can draw the following conclusions:

> **Observation 1:** *LVLMs can produce significantly different reasoning lengths with modest changes in accuracy via length rewards, suggesting potential for balancing reasoning depth and performance.*

## 3.2 Data Distribution Analysis

Considering that overthinking models tend to generate verbose reasoning responses regardless of question difficulty, we next investigated how data distribution—particularly the presence of samples with varying difficulty levels—might naturally influence reasoning length and performance.

To this end, we stratified the Geometry3K training dataset into three difficulty tiers using the pass@8 metric (i.e., the probability of correctly solving a question within eight attempts): Easy ($0.75 \leq pass@8$), Medium ($0.25 < pass@8 < 0.75$), and Hard ($pass@8 \leq 0.25$). This catego-

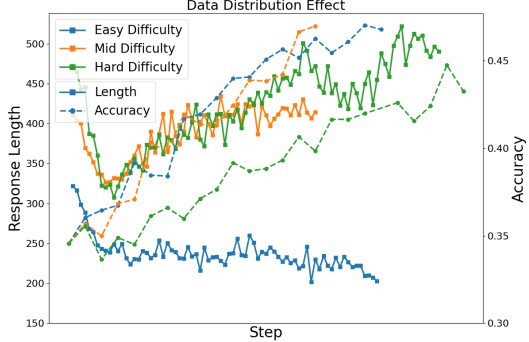

Figure 3: Effect of data distribution, especially difficulty on reasoning length and accuracy.

rization resulted in approximately 35% Easy, 25% Medium, and 40% Hard samples.

Figure 3 illustrates how training on these different difficulty distributions affected model behavior. Models trained exclusively on Hard samples generated significantly longer responses but showed only

marginal accuracy improvements. In contrast, training on Easy samples produced shorter responses while improving accuracy. Models trained on Medium samples showed modest length increase and the highest accuracy. Based on the findings above, we can draw the following conclusions:

> **Observation 2:** *Question difficulty acts as an implicit regulator of reasoning length, suggesting that data distribution can be strategically leveraged to achieve adaptive fast-slow thinking.*

## 4 Fast-Slow Thinking GRPO

Building upon the aforementioned observations, we begin by introducing two complementary metrics to quantify the difficulty of multimodal questions, which facilitates dynamic data selection during reinforcement learning(§4.1). Subsequently, we introduce FAST-GRPO, a variant of GRPO specifically designed to balance fast and slow reasoning by leveraging adaptive length-based rewards conditioned on question difficulty(§4.2).

### 4.1 Multimodal Question Difficulty Estimation

Given our findings that data distribution influences both reasoning length and performance, accurately gauging data difficulty becomes essential for making dynamic adjustments to data distribution. To address this need, we propose two complementary metrics to measure the difficulty of a given question for the policy model: one directly evaluates the intrinsic difficulty of the multimodal question itself, while the other measures its difficulty relative to the policy model's current capabilities.

**Extrinsic Difficulty.** We first quantify question difficulty relative to the policy model through the empirical success rate $S_{\text{extrinsic}} = 1 - \text{pass@k}$ , where $\text{pass@k} = c/k$ represents the proportion of correct solutions among $k$ rollouts. This metric is computed online, reflecting the model's evolving capabilities.

**Intrinsic Difficulty.** While extrinsic difficulty reflects a model's ability to solve problems, it may not fully capture the inherent visual complexity of the questions. We therefore introduce ***image complexity*** as an indicator that specifically evaluates the intrinsic difficulty within questions. Specifically, we use the Gray-Level Co-occurrence Matrix (GLCM) score, which analyzes how frequently pairs of pixels with specific intensity values occur at defined spatial relationships [34, 35]. However, GLCM captures only low-level image complexity based on pixel-level interactions and fails to account for higher-level semantic information. We therefore additionally employ the ViT classification entropy based on the output of the feature layer [36, 37] for image semantics complexity, providing a high-level representation of conceptual difficulty. The image complexity is computed as follows:

$$H_{image} = -\frac{1}{P} \sum_{p=1}^{P} H(g_p) - H(v) \tag{2}$$

where $g_p$ represents the GLCM for image patch $p$, $H(g_p)$ is the entropy of the co-occurrence probabilities across multiple radii and orientations; $v$ denotes the feature output from the final layer of the ViT classifier, and $H(v) = -\sum_{j}^{N} p_j \log p_j$ is the entropy of the predicted probability distribution over $N$ classes, with $p_j$ being the probability for class $j$.

We integrate these metrics to get a comprehensive difficulty estimation ($S_{\text{difficulty}}$) of a multimodal question. Questions with higher difficulty scores typically correspond to greater visual complexity and greater extrinsic difficulty for the policy model.

$$S_{\text{difficulty}} = S_{\text{extrinsic}} \cdot H_{image} \tag{3}$$

**Slow-to-Fast Sampling.** We adopt curriculum strategies for the fast–slow thinking paradigm, using online-computed difficulty metrics. Two variants are considered: (i) **Binary**: distinct training phases. In *early epochs*, exclude easy samples ($S_{\text{extrinsic}} \leq 0.25$) to strengthen reasoning on hard

questions; in *later epochs*, exclude hard samples ($S_{\text{extrinsic}} \geq 0.75$) to practice concise reasoning. (ii) **Continuous**: smoothly shift sampling probability from harder to easier questions over epochs, enabling a gradual transition from slow to fast thinking. Binary enforces a clear capability-efficiency separation, while Continuous offers a gentler progression.

## 4.2 FAST-GRPO

With the carefully designed difficulty estimation methods for multimodal question, we introduce **FAST-GRPO**, a tailored variant of GRPO that balances fast and slow reasoning by incorporating adaptive length-based rewards conditioned on question difficulty, as illustrated in Algorithm 1.

**Difficulty-Aware Length Reward Shaping.** In addition to the accuracy reward $r_a$ and format reward $r_f$, we propose a difficulty-aware length reward $r_t$ as follows, which guides the model to employ the appropriate reasoning approach based on question difficulty.

$$r_t = \begin{cases} 1 - \frac{L}{L_{avg}} & \text{if } (S_{\text{d}} < \theta) \wedge (r_a = 1) \\ min(\frac{L}{L_{avg}} - 1, 1) & \text{if } (\theta \leq S_{\text{d}}) \wedge (r_a = 0) \\ 0 & \text{otherwise} \end{cases}$$

(4)

where $S_{\text{d}}$ is the difficulty score, $\theta$ is the 80th percentile difficulty threshold across the batch, and $L_{avg}$ is the average length computed in the batch of responses. For less complex questions ($S_{\text{d}} < \theta$), the reward encourages fast thinking for correct trajectories, specifically rewards trajectories towards shorter than average length. Conversely, for complex questions, the reward encourages thorough reasoning for incorrect trajectories. Importantly, this reward is capped at 1, preventing excessive verbosity even for complex problems. Difficulty threshold $\theta$ is a hyperparameter, for which we analyse sensitivity in §5.4.

Following Deepseek-R1 setting [8, 19], we

---

**Algorithm 1** FAST-GRPO Training

**Require:** Base model $\pi_\theta$, selected dataset $\mathcal{D}$, image complexity $H_{img}$
1: Initialize $\pi_{\text{ref}} \leftarrow \pi_\theta$
2: **for** epoch = 1 **to** $N$ **do**
3:      Sample batch $\{q_i\} \sim \mathcal{D}$
4:      Generate $\{o_i^j\}_{j=1}^G \sim \pi_\theta(q_i)$
5:      Compute:
        •   $S_{ext} = 1 - \text{pass@k}(q_i)$
        •   $S_d = S_{ext} \cdot H_{img}$
        •   $\beta_i = \beta_{\min} + (\beta_{\max} - \beta_{\min})(1 - S_{ext})$
6:      Filter samples via Slow-to-Fast Sampling
7:      Compute rewards: $r_i^j = r_a + \lambda_f r_f + \lambda_t r_t$

$$r_t = \begin{cases} 1 - \frac{L}{L_{avg}} & \text{if } (S_d < \theta) \wedge (r_a = 1) \\ min(\frac{L}{L_{avg}} - 1, 1) & \text{if } (S_d \geq \theta) \wedge (r_a = 0) \\ 0 & \text{otherwise} \end{cases}$$

8:      Update policy:

$$\max_\theta \mathbb{E} \left[ \text{clip}\left(\frac{\pi_\theta}{\pi_{\text{old}}}\right) \hat{A}^j - \beta_i D_{\text{KL}}(\pi_\theta \| \pi_{\text{ref}}) \right]$$

9: **end for**

---

define the final reward function as a linear combination of these components: $r_i = r_a + \lambda_f r_f + \lambda_t r_t$. This difficulty-aware length reward necessitates encouraging exploration for complex problems while maintaining efficient, accurate responses for simpler ones.

**Difficulty-Aware KL Regularization.** In addition to the aforementioned length reward that encourages adaptive response length for questions of varying difficulty, the KL divergence term constrains the policy model's deviation from the reference model to achieve an exploitation-exploration balance, which impacts learning effectiveness across questions of different difficulty levels [6, 31]. Our KL coefficient sensitivity analysis in § 5.4 also reveals that no single static $\beta$ value optimally serves questions across difficulty levels. Lower KL constraints benefit challenging questions by enabling broader exploration, while stronger regularization maintains performance on simpler tasks. To address this issue, we implement a dynamic coefficient $\beta_d$ for difficulty-aware regularization.

$$\beta_d = \beta_{\min} + (\beta_{\max} - \beta_{\min}) \cdot (1 - S_{\text{extrinsic}}) \tag{5}$$

We give a simple theoretical analysis to demonstrate how difficulty-aware $\beta_d$ enhances learning on varying questions by decomposing the gradient coefficient from the gradient Equation 6:

$$\nabla_\theta \mathcal{J}_{GRPO}(\theta) = \mathbb{E}_{[q \sim P(Q), \{o_i\}_{i=1}^G \sim \pi_{\theta_{old}}(O|q)]} \left[ \frac{1}{G} \sum_{i=1}^G \frac{1}{|o_i|} \sum_{t=1}^{|o_i|} [GC_{GRPO}(q,o)] \nabla_\theta \log \pi_\theta(o_{i,t}|q, o_{i,<t}) \right]$$

(6)

$$GC_{GRPO}(q,o) = \underbrace{\hat{A}_i}_{\text{Advantage Signal}} + \underbrace{\beta_d \left( \frac{\pi_{\text{ref}}(o_i|q)}{\pi_\theta(o_i|q)} - 1 \right)}_{\text{Adaptive KL Regularization}} \tag{7}$$

As shown in Equation 7, the gradient coefficient consists of the advantage signal driving policy improvement and the adaptive KL regularization term. For high-difficulty questions, $\beta_d$ approaches $\beta_{\min}$, weakening the KL regularization and allowing the policy update to be dominated by the advantage signal. For low-difficulty questions, $\beta_d$ approaches $\beta_{\max}$, restricting policy deviation to ensure stability. Besides, the length normalization term $\frac{1}{|o_i|}$ explicitly affects gradient updates, providing theory insight into Observation 2: for incorrect responses, it encourages longer outputs by reducing per-token penalties, while for correct responses, it encourages brevity through stronger per-token updates. This creates an implicit bias toward increasing response length for difficult questions where models generate more incorrect rollouts, while naturally promoting shorter responses for simpler questions that yield more correct solutions.

## 5 Experiments

In this section, we evaluate the efficacy of our method for LVLM reasoning.

### 5.1 Experimental Setup

**Training Dataset.** Starting with 500K questions from LLaVA-CoT [23], Mulberry [39], and MathV-360K [40], we first apply filters for answer verifiability. We deduplicate questions, retain only rule-based verifiable answers [41], and standardize to closed-form questions (e.g., multiple-choice, numeric answers). Second, we apply Slow-to-Fast sampling to remove questions with extreme extrinsic difficulty scores ($S_{\mathrm{extrinsic}} = 0$ or $1$), yielding 18K training questions. We display its distribution in Figure 6 and the specific source in the appendix.

**Evaluation Benchmarks.** we evaluate on 7 widely used multimodal benchmarks: (1) Math-Vision [42], (2) MathVerse [43], (3) Math-Vista [44], (4) MM-Math [45], (5) WeMath [46], (6) DynaMath [47], and (7) MM-Vet [48]. The first six cover various mathematical reasoning tasks, while MM-Vet examines general multimodal abilities. We report both accuracy and response length for all benchmarks. We also conduct additional cross-domain evaluations, including science reasoning (MM-K12 [22]), open-domain VQA (Bingo [49], MMHAL [50]), low-level visual perception (MMVP [51]), and comprehensive calibrated evaluation (MMEval-Pro [52]). Details are provided in Appendix K.

Table 2: Comparison of different training methods and training samples.

| Method | Training Stage | | | |
|--------|------|--------|-----|--------|
| | SFT | Sample | RL | Sample |
| Virgo [38] | ✓ | 5K | ✗ | – |
| Mulberry [39] | ✓ | 260K | ✗ | – |
| LMM-R1 [17] | ✗ | – | ✓ | 105K |
| MM-R1 [21] | ✗ | – | ✓ | 6K |
| MM-Eureka [22] | ✗ | – | ✓ | 56K |
| Curr-ReFT [18] | ✓ | 1.5K | ✓ | 9K |
| OpenVLThinker [10] | ✓ | 35K | ✓ | 15K |
| Vision-R1 [15] | ✓ | 200K | ✓ | 10K |
| R1-OneVision [16] | ✓ | 155K | ✓ | 10K |
| **FAST (Ours)** | ✗ | – | ✓ | 18K |

**Baselines.** For slow thinking reasoning, we compare with three categories of approaches: (1) SFT on distilled data (LLaVA-CoT, Mulberry, Virgo); (2) RL-only training (MM-Eureka, LMM-R1, MM-R1); and (3) Two-stage approaches combining SFT and RL (R1-OneVision, Curr-ReFT, OpenVLThinker, Vision-R1). A comparative analysis of training methodologies and samples across these baselines is presented in Table 2. For fast-slow thinking comparison, we evaluate against fast thinking methods using various reward shaping techniques: Kimi 1.5's length penalty [25], cosine function rewards [26], and DAST [27]. Table 10 details these different reward formulations.

### 5.2 Main Results

We report the main results concerning reasoning accuracy and reasoning length.

**Reasoning Accuracy.** Table 4 reports the main results of reasoning performance. First, FAST achieves state-of-the-art results on MathVista with 73.8 and MathVerse with 50.6, outperforming leading-edge closed-source LVLMs like GPT-4o. Second, on more challenging benchmarks, MathVision and MM-Math, FAST achieves competitive results, validating FAST's ability to solve complex questions. Third, FAST improves Qwen2.5-VL-7B, our base model, with an average accuracy improvement of over 10%. Fourth, FAST improves Qwen2.5-VL-3B with an average accuracy

Table 3: Main results on reasoning benchmarks compared with slow-thinking methods. For each benchmark, we report both accuracy (acc.) and response length (len.). Tokens are counted with Qwen2.5-VL's tokenizer.

| Method | MathVision | | MathVerse | | MathVista | | MM-Math | | WeMath | | DynaMath | | MM-Vet | |
|---|---|---|---|---|---|---|---|---|---|---|---|---|---|---|
| | Acc. | Len. | Acc. | Len. | Acc. | Len. | Acc. | Len. | Acc. | Len. | Acc. | Len. | Acc. | Len. |
| *Closed-Source Model* | | | | | | | | | | | | | | |
| GPT-4o | 30.4 | – | 49.9 | – | 63.8 | – | 31.8 | – | 69.0 | – | 63.7 | – | 80.8 | – |
| Claude-3.5 Sonnet | 37.9 | – | 46.3 | – | 67.7 | – | – | – | – | – | 64.8 | – | 68.7 | – |
| Qwen-VL-Max | 39.3 | – | 47.3 | – | 74.2 | – | 45.6 | – | – | – | – | – | 73.2 | – |
| MM-Eureka | 26.9 | – | 40.4 | – | 67.1 | – | – | – | – | – | – | – | 60.7 | – |
| LLaVA-CoT | 16.4 | – | 20.3 | – | 54.8 | – | 22.6 | – | – | – | 44.8 | – | 60.3 | – |
| *Base Qwen2-VL-7B* | | | | | | | | | | | | | | |
| Qwen2-VL-7B | 18.8 | 443.0 | 31.9 | 388.9 | 58.2 | 265.9 | 20.2 | 661.7 | 50.5 | 294.3 | 39.8 | 298.4 | 62.0 | 132.5 |
| Mulberry | 23.4 | 349.2 | 39.5 | 364.3 | 62.1 | 275.0 | 23.7 | 467.0 | 50.4 | 372.1 | 46.8 | 273.3 | 43.9 | 218.3 |
| Virgo | 24.0 | – | 36.7 | – | – | – | – | – | – | – | – | – | – | – |
| *Base Qwen2.5-VL-3B* | | | | | | | | | | | | | | |
| Qwen2.5-VL-3B | 21.2 | 450.6 | 34.6 | 362.3 | 62.3 | 212.9 | 33.1 | 627.9 | 50.4 | 323.7 | 48.2 | 270.9 | 61.3 | 138.8 |
| Curr-ReFT | 20.1 | 240.1 | 36.3 | 121.6 | 61.9 | 95.9 | 28.6 | 301.5 | 57.3 | 156.0 | 43.8 | 146.4 | 62.0 | 117.6 |
| LMM-R1 | 25.2 | 447.8 | 41.8 | 423.9 | 63.2 | 245.0 | 36.5 | 634.5 | 62.9 | 382.5 | 53.1 | 341.6 | 65.9 | 166.3 |
| **FAST-3B (Ours)** | 26.8 | 323.5 | 43.0 | 286.3 | 66.2 | 158.7 | 39.4 | 425.0 | 63.1 | 244.9 | 54.4 | 213.7 | 64.0 | 112.7 |
| *Base Qwen2.5-VL-7B* | | | | | | | | | | | | | | |
| Qwen2.5-VL-7B | 25.6 | 443.0 | 46.9 | 388.9 | 68.2 | 189.1 | 34.1 | 666.7 | 61.0 | 294.3 | 58.0 | 273.3 | 67.1 | 132.5 |
| MM-R1 | 30.2 | 324.6 | 49.8 | 283.9 | 71.0 | 185.6 | 41.9 | 528.5 | 67.9 | 235.7 | 57.5 | 254.2 | 70.6 | 137.9 |
| Vision-R1 | – | – | 52.4 | – | 73.5 | – | 40.4 | – | – | – | – | – | – | – |
| R1-OneVision | 29.9 | 692.8 | 46.4 | 631.5 | 64.1 | 402.5 | 34.1 | 688.6 | 61.8 | 591.9 | 53.5 | 560.6 | 71.6 | 440.7 |
| OpenVLThinker | 29.6 | 457.2 | 47.9 | 398.4 | 70.2 | 305.7 | 33.1 | 549.7 | 64.5 | 326.7 | 57.4 | 382.1 | 68.5 | 312.7 |
| **FAST-7B (Ours)** | 30.6 | 204.8 | 50.6 | 201.0 | 73.8 | 120.7 | 44.3 | 335.6 | 68.8 | 170.3 | 58.3 | 164.8 | 71.2 | 114.1 |

Table 4: Main results of accuracy and length compared with fast-thinking reward shaping methods.

| Method | MathV | | MathVista | | MathVer. | | WeMath | | MM-Vet | | Avg. | |
|---|---|---|---|---|---|---|---|---|---|---|---|---|
| | Acc. | Len. | Acc. | Len. | Acc. | Len. | Acc. | Len. | Acc. | Len. | Acc. | Len. |
| Kimi | 25.9 | 78.9 | 71.1 | 58.1 | 48.2 | 105.8 | 66.2 | 75.3 | 67.1 | 57.1 | 55.7 | 75.0 |
| CosFn | 27.9 | 396.4 | 72.1 | 247.2 | 49.6 | 383.9 | 68.1 | 311.9 | 71.1 | 148.9 | 57.8 | 297.7 |
| DAST | 27.0 | 281.1 | 72.9 | 93.5 | 48.5 | 194.5 | 67.4 | 148.9 | 67.6 | 66.3 | 56.7 | 156.9 |
| **FAST** | 30.6 | 204.8 | 73.8 | 120.7 | 50.6 | 201.0 | 68.8 | 170.3 | 71.2 | 114.1 | 59.0 | 162.2 |

improvement of over 14%, demonstrating that our method can be applied to different-sized models. Further scalability results on a 32B-parameter model are provided in Appendix J. Lastly, FAST maintains its general multimodal ability, evidenced by improved performance on MM-Vet, and further demonstrates strong generalization beyond math-centric benchmarks in science reasoning and open-domain VQA. In these evaluations, FAST improves its base model by 7–9% in physics, chemistry, and biology, and on open-domain VQA matches or surpasses strong baselines, showing effectiveness across diverse reasoning domains. Detailed results are provided in Appendix K.

**Reasoning Length.** Tables 3 and 4 report the main results of reasoning length. First, FAST achieves a significant reduction of average reasoning length compared to slow thinking methods, from 32.7% against MM-R1 to 67.3% versus R1-OneVision, while preserving comparable or better reasoning accuracy. Second, compared to other fast thinking methods in LLMs, FAST achieves a modest reasoning length reduction and better reasoning accuracy. Third, FAST achieves slower thinking on more challenging questions, producing 60% longer responses on Hard than Easy of Geometry 3K as shown in Table 1 and averaging 79% more tokens on MM-Math. Lastly, in cross-domain evaluations ( §K), FAST yields substantially shorter responses: on MM-K12, average length drops by ∼106 tokens (33.8%) vs. its base model and over 30% vs. strong slow-thinking baselines. In open-domain VQA (Bingo, MMHal), outputs are consistently 15–25% shorter, showing effective control of reasoning length beyond math tasks.

## 5.3 Ablations

We conduct ablation studies to validate the effectiveness of each design of our method: Data Sampling, thinking reward shaping, and difficulty-aware optimization. The results are represented in Table 5. We can draw the following conclusions. First, without Data Sampling, reasoning accuracy seriously degrades on all benchmarks, highlighting the critical role of proper data distribution. Second, our thinking reward significantly reduces relative 42% response length with minor reasoning accuracy degradation, from 31.5 to 30.6 on MathVision. Third,

Table 5: Ablation Results on MathVista, MathVision, and MathVerse. More details of naive GRPO refer to appendix § F.

| Model | MathVista | MathV. | MathVer. | Len. |
|---|---|---|---|---|
| Qwen-2.5-VL-7B | 68.2 | 25.6 | 46.9 | 340.3 |
| FAST | 73.8 | 30.6 | 50.6 | 175.5 |
| w/o Data Sampling | 69.9 | 27.2 | 48.4 | 257.3 |
| w/o Thinking Reward | 73.6 | 31.5 | 45.9 | 302.2 |
| w/o Difficulty Aware | 72.0 | 29.5 | 49.2 | 171.6 |
| Naive GRPO | 67.2 | 25.3 | 47.6 | 205.4 |
| + *early stop* | 70.4 | 28.1 | 48.9 | 243.6 |

the difficulty-aware regularization demonstrates robust improvement across all benchmarks, with a 1.8-point absolute increase on MathVista.

## 5.4 Analyses

**Effect of Slow to Fast Sampling.** We further investigate the effect of Slow to Fast sampling by comparing our Slow to Fast sampling with alternative approaches: Fast to Slow, i.e., excluding hard samples early, easy samples later, and Dynamic Sampling [53], i.e., always filtering out Easy and Hard samples). As shown in Table 6, Fast to Slow yields comparable accuracy but shows degradation on challenging MathVision, while Dynamic Sampling leads to 80%

Table 6: Results on the effect of Slow-to-Fast Sampling.

| Method | MathV. | MathVista | MathVer. | Len. |
|---|---|---|---|---|
| No Selection | 25.3 | 67.2 | 47.6 | 205.4 |
| Dynamic Sampling | 27.0 | 73.2 | 50.3 | 317.9 |
| Fast to Slow | 26.3 | 72.9 | 50.2 | 266.1 |
| Continuous Slow-to-Fast | 30.9 | 74.4 | 51.0 | 221.2 |
| Binary Slow-to-Fast | 30.6 | 73.8 | 50.6 | 175.5 |

longer responses without better accuracy improvements. We also compared our binary Slow-to-Fast sampling against a continuous variant to examine the effect of gradual curriculum shifts. This additional comparison is reported in Appendix I.

Table 7: Results on the effect of SFT vs GRPO.

| Samples | Annotator | MathV. | MathVis. | MathVer. |
|---|---|---|---|---|
| 260K | 4o | 27.9 | 64.0 | 46.5 |
| 200K | R1 | 18.8 | 66.8 | 47.1 |
| 18K | – | 30.6 | 73.8 | 50.6 |

Table 8: Correlation between image complexity metrics and human judgments.

| Metric | SRCC | PLCC |
|---|---|---|
| GLCM entropy score | 0.75 | 0.77 |
| $H_{img}$ | 0.49 | 0.54 |

**Effect of SFT versus GRPO.** As shown in Table 7, to further verify the efficacy of our FAST compared to SFT methods, we compare our method with SFT using: (1) 260K structured CoT data from GPT-4o [39] and (2) 200K long CoT from Deepseek-R1 [15]. SFT on Deepseek-R1 data produces overthinking responses with degraded reasoning, while SFT on GPT-4o data mimics fixed structures without substantial gains. In contrast, FAST with just 18K samples demonstrates superior performance across all benchmarks.

**Validation of Image Complexity.** In our image complexity design, we utilize the GLCM entropy score [34, 35] to measure texture complexity and ViT classification entropy [36, 37] for semantic complexity. Zhang et al. [34] demonstrated that GLCM entropy achieves strong human alignment with a Spearman Rank-order Correlation Coefficient (SRCC) of 0.75 and a Pearson Linear Correlation Coefficient (PLCC) of 0.77. To validate the effectiveness of our combined metric $H_{img}$ on our specific dataset, we followed the methodology in [34], having three participants rate 200 sampled training images on a 5-point scale based on visual detail complexity. As shown in Table 8, while our combined metric $H_{img}$ demonstrates moderate correlation with human judgments (SRCC=0.49, PLCC=0.54), it maintains well alignment with human perception.

**Difficulty Threshold Analysis.** We use the 80th percentile of batch difficulty as our threshold $\theta$. Figure 4 shows our grid search results: the 100th percentile yields concise responses (140.8 tokens) but reduces accuracy, while a 0 threshold produces excessive verbosity (486.2 tokens) with

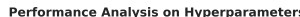

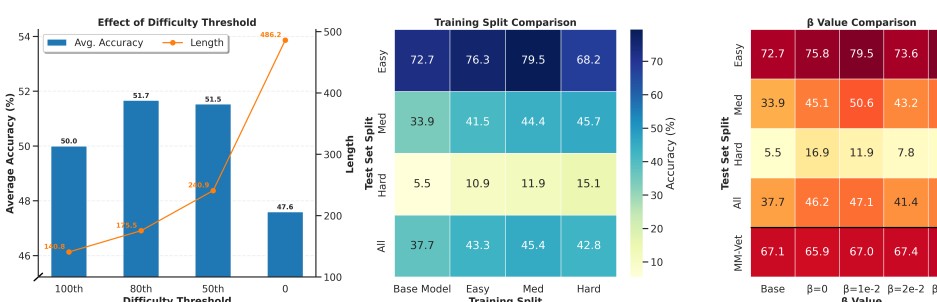

Figure 4: **Left:** Results on the effect of difficulty threshold. The average accuracy is computed across MathVision, MathVerse, and MathVista. **Middle:** Test set results with different difficulty level training split comparisons on Geometry 3K. **Right:** Test set results with different $\beta$ value comparison in pilot experiments on Geometry 3K. OOD results comparison on MM-Vet benchmark.

performance degradation. The 50th percentile achieves comparable accuracy to our 80th percentile but with 37% longer responses, confirming our choice effectively balances accuracy and conciseness.

**Extrinsic Difficulty Analysis.** Figure 4 reveals how training on different extrinsic difficulty splits affects model performance. First, training on Medium difficulty samples yields the best overall performance (45.4%), providing optimal balance for learning. Second, we observe clear difficulty-specific transfer effects: Easy training improves Easy test performance (76.3%), Medium training benefits Medium tests (44.4%), and Hard training significantly boosts Hard test performance (15.1% vs. base 5.5%). However, Hard sample training degrades Easy performance (68.2% vs. base 72.7%), while Easy training shows limited transfer to Hard problems. These findings support our Slow-to-Fast sampling strategy, demonstrating that no single difficulty level is optimal for all test cases.

**KL Coefficient Analysis.** Recent works [53, 54] suggest that removing KL constraints can enhance long-form reasoning in language models. We explored this effect in visual reasoning through a grid search on the KL coefficient $\beta$ (Figure 4). Our analysis reveals that lower $\beta$ values significantly improve performance on Hard questions (16.9% at $\beta = 0$ vs. base 5.5%) by enabling greater exploration, but risk catastrophic forgetting on previously mastered tasks. Conversely, higher $\beta$ values maintain strong performance on Easy questions and improve out-of-distribution generalization (69.2% at $\beta = 5e - 2$ on MM-Vet), but restrict exploration on complex reasoning tasks. These findings demonstrate no static $\beta$ value optimally serves questions across all difficulty levels—Hard questions benefit from looser constraints while Easy ones and generalization require stronger regularization.

**In-depth Analysis of Multiplying Estimated Difficulties.** The multiplicative form $S_{\text{difficulty}} = S_{\text{extrinsic}} \cdot H_{\text{image}}$ jointly captures empirical hardness and intrinsic visual complexity. One theoretical concern is that this product could give low scores for cases that are hard for the model but visually simple, potentially leading to fast-thinking behaviour when slow reasoning is needed. In practice, such mismatches are rare (less than 5% of our training set), and our reward design in §4.2 assigns zero length reward in these cases, avoiding contradictory signals. We also tested a weighted-sum alternative $S_{\text{difficulty}}^{(\text{sum})} = \alpha S_{\text{extrinsic}} + (1 - \alpha)H_{\text{image}}$, where $\alpha = 0.5$, and found almost identical performance to the multiplicative form across MathVista, MathVision, and MathVerse (differences within 0.5% accuracy). These results confirm that FAST's difficulty estimation is robust to this potential corner case and to the choice of combination strategy. Details are provided in §H.

## 5.5 Case Studies and Failure Mode Analysis

We complement our quantitative results with qualitative illustrations of FAST's behaviour. Appendix §L provides examples where FAST adapts its reasoning, from concise answers on simple problems to expanded chains on complex ones, as well as typical failure cases. Here, we focus on a systematic analysis of these failures.

To understand *when* and *why* FAST-GRPO succeeds or fails, we analyse all incorrect responses from FAST-7B, R1-OneVision-7B, and the base model (Qwen2.5-VL-7B) on MATHVISTA. We

observe three recurring failure patterns (Figure 5): (i) **Visual Perception Failures** — where the model incorrectly extracts or interprets visual cues (e.g., scales, chart values, spatial relations); (ii) **Reasoning Error Propagation** — where a mid-chain mistake contaminates subsequent logical steps; and (iii) **Knowledge Conflict & Gap** — where language priors override contradictory visual evidence, or the model hallucinates in the absence of domain knowledge.

**Key Insights.** First, adaptive fast–slow thinking substantially reduces reasoning-related failures: FAST-7B cuts *Reasoning Error Propagation* and *Knowledge Conflict* cases by ~27% and ~19% relative to its base model. Shorter, targeted reasoning chains leave fewer opportunities for mid-proof errors and help suppress hallucinations caused by overextended thought.

Second, perception, not reasoning, is the dominant bottleneck: over half of FAST-7B's errors stem from visual misinterpretation. Once a spatial relation is mis-localised or a key numeric value misread, even perfectly structured reasoning will converge to an incorrect answer. Future

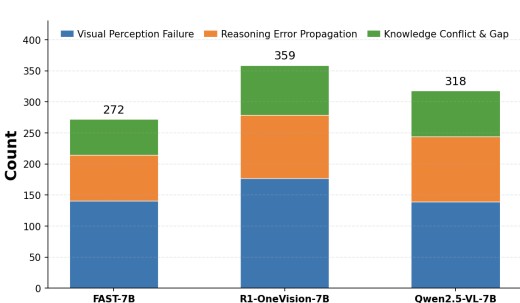

Figure 5: Error breakdown by category.

gains will likely come from strengthening the *input stage*, e.g., fine-grained OCR, calibrated scale reading, robust chart and graph value extraction, and accurate spatial grounding, so adaptive reasoning can operate on correct evidence.

# 6   Related Work

We review approaches for LVLM reasoning and methods addressing overthinking in LLMs.

**Slow-thinking methods for LVLMs. SFT-RL two-stage** methods leverage high-quality reasoning trajectories while inadvertently behavior cloning overthinking. Examples include Mulberry [39, 55] using MCTS from GPT-4o, LLaVA-CoT [23] with structured reasoning stages, and Virgo [38] finetuning on text-only reasoning chains. Vision-R1 [15], R1-OneVision [16], and OpenVLThinker [10] first collect distilled data from advanced models before applying SFT and RL. **RL-only** methods directly employ RL to improve reasoning accuracy but struggle with scaling response length [56, 21, 22]. Visual-RFT [56] uses GRPO for various vision tasks, while MM-R1 [21], LMM-R1 [17], and MM-Eureka [22] apply RL on base models with curated visual reasoning questions.

**Fast-Slow thinking methods for LLMs.** Methods addressing overthinking in LLMs include **inference-stage** approaches include TALE [30] enforcing token budgets in prompt and CCoT [29] providing concise examples in context. **Training-stage** approaches include O1-Pruner [28] using a tailored RL objective to reduce verbosity, CoT-Value [57] fine-tuning on varied-length reasoning chains to learn dynamic thinking, and Kimi [25] proposing length penalty rewards in RL and Long2Short DPO [58] to shorten length. DAST [27] and CosineReward [26] encourage shorter correct responses and longer in- correct responses via curated length rewards. While effective for text-only tasks, these approaches remain largely unexplored for LVLM reasoning.

# 7   Conclusion

We presented FAST, a framework enabling LVLMs to dynamically adapt reasoning depth based on question characteristics, addressing the overthinking phenomenon. Through empirical analysis, we developed FAST-GRPO with three components: model-based metrics for question characterization, adaptive thinking rewards, and difficulty-aware KL regularization. Extensive experiments demonstrated that FAST achieves state-of-the-art accuracy with over 10% improvement compared to the base model while reducing token usage compared to previous slow-thinking approaches, effectively balancing reasoning length and accuracy.

## 8 Acknowledgement

This work was supported in part by the "Pioneer" and "Leading Goose" R&D Program of Zhejiang (No. 2025C02037), the Earth System Big Data Platform of the School of Earth Sciences, Zhejiang University, Alibaba-Zhejiang University Joint Research Institute of Frontier Technologies, and the Science and Technology Project of State Grid Beijing Electric Power Company (Project Title: Research on Urban Cable Network Operation Status Detection and Risk Identification Technology Based on Soft Robots and Artificial Intelligence, Project Number: 520246250003).

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

This Appendix for *"Fast-Slow Thinking GRPO for Large Vision-Language Model Reasoning"* is organized as follows:

- **Experimental Setup and Reproducibility.** In §B we detail the implementation settings; §C describes the training and evaluation datasets; §D compares different length reward shaping methods; §E provides the human evaluation prompt for image complexity.
- **Additional Experimental Analyses.** §F analyses naive GRPO behaviors; §G reports statistical significance of main results; §L presents case studies and failure mode examples; §K gives cross-domain evaluation results (science reasoning, open-domain VQA, low-level visual perception, calibrated evaluation); §J shows scalability experiments on a 32B-parameter LVLM.
- **Discussion and Limitations.** §A discusses potential limitations of FAST-GRPO and directions for future work.

## A  Limitations

While our FAST framework demonstrates significant improvements in balancing reasoning length and accuracy, we acknowledge several limitations in our current work. Due to computational resource constraints, we were only able to evaluate our approach on models up to 32B parameters (Qwen2.5-VL-32B). The effectiveness of fast-slow thinking mechanisms may scale differently with larger models (e.g., models with 70B+ parameters), which could potentially exhibit different reasoning patterns and overthinking behaviors.

## B  Implementation Details

We implement FAST using Qwen2.5-VL-3B and 7B as our base models. Below we detail our training setup and hyperparameters.

**General Training Hyperparameters.** For FAST training, we use our 18K dataset with a learning rate of 1e-6, a batch size of 512. We set the maximum sequence length to 4096 for both prompts and generation, and apply BF16 precision throughout training. The training process runs for 10 epochs, requiring approximately 600 H100 GPU hours. We use the prompt: *You FIRST think about the reasoning process as an internal monologue and then provide the final answer. The reasoning process MUST BE enclosed within $<think></think>$ tags. The final answer MUST BE put in $<answer></answer>$ tags.*

Table 9: Training Hyperparameters

| Hyperparameter | Value |
|---|---|
| Model | Qwen2.5-VL |
| Epochs | 10 |
| Learning Rate | 1e-6 |
| Train Batch Size | 512 |
| Temperature | 1.0 |
| Rollout per Prompt | 8 |
| Prompt Max Length | 4096 |
| Generation Max Length | 4096 |
| Max KL Coefficient | 0.03 |
| Min KL Coefficient | 0.001 |
| Precision | BF16 |
| Max Pixels | 1000000 |
| $\lambda_f$ | 0.5 |
| $\lambda_t$ | 0.5 |
| Difficulty | $\begin{cases} \text{Easy} & \text{if } 0.75 \le \text{pass@k} \\ \text{Hard} & \text{if pass@k} \le 0.25 \\ \text{Medium} & \text{otherwise} \end{cases}$ |
| Difficulty Threshold | 80th percentile |

**Method-specific Training Hyperparameters.** For our reinforcement learning approach, we employ a temperature of 1.0, 8 rollouts per question, and a KL coefficient ranging from 0.001 (min) to 0.03 (max). The reward weighting factors are set to 0.5. The difficulty threshold is set at the 80th percentile. For GLCM computation, following prior setting [34], $g_p$ is derived from local patch $p$ in the original image with 64 gray levels, defined by radius $\delta = [1, 2, 3, 4]$ and orientation $\theta = [0°, 45°, 90°, 135°]$. In practice, we divide the gray image into local patches of size 64.

**Computation Environment.** All training experiments were conducted using H20 GPUs. Model inference in evaluations is performed using the vLLM framework [59], and our training implementation extends the VeRL codebase [60].

The complete set of hyperparameters is provided in Table 9. We commit to releasing all the code, data, and model checkpoints for experimental results reproducibility.

## C   Datasets

Our training dataset comprises samples from four main categories: (1) *Mathematical* problems, including data from MathV360K, Geometry3K, and other mathematical reasoning datasets; (2) *Visual QA* tasks, sourced from ShareGPT4V, Vizwiz, and additional visual question answering benchmarks; (3) *Science* problems from AI2D, ScienceQA, and other scientific reasoning datasets; and (4) *Figure Understanding* tasks from DocVQA, ChartQA, and other document and chart comprehension datasets. The distribution is balanced across these categories, with Mathematical problems constituting the largest portion, followed by Figure Understanding, Science, and Visual QA tasks.

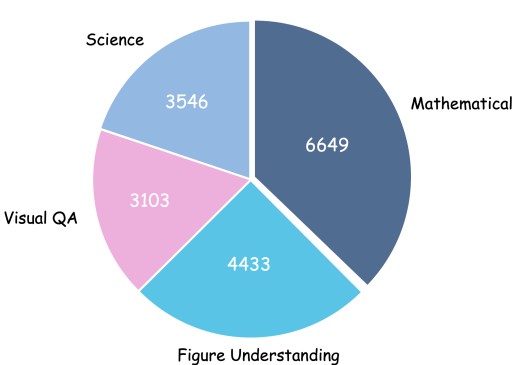

Figure 6: Distribution of Training Dataset Sources by Category.

## D   Length Rewards

We provide a comparison of different length rewards in Talbe 10.

Table 10: Comparison of different length reward shaping methods.

| Method | Length Reward |
|---|---|
| Kimi Length Penalty [25] | $\begin{cases} 0.5 - \frac{\text{len}(i) - \text{min\_len}}{\text{max\_len} - \text{min\_len}} & \text{if correct} \\ \min(0, 0.5 - \frac{\text{len}(i) - \text{min\_len}}{\text{max\_len} - \text{min\_len}}) & \text{otherwise} \end{cases}$ |
| CosFn [26] | $\eta_{min} + \frac{1}{2}(\eta_{max} - \eta_{min})(1 + \cos(\frac{t\pi}{T}))$ 
 where $t$ is generation length, $T$ is maximum length 
 $\eta_{min}/\eta_{max}$ are min/max rewards 
 For correct answers: $\eta_{min} = r_0^c, \eta_{max} = r_L^c$ 
 For wrong answers: $\eta_{min} = r_0^w, \eta_{max} = r_L^w$ |
| DAST [27] | $\begin{cases} \max(-0.5\lambda + 0.5, 0.1) & \text{if correct} \\ \min(0.9\lambda - 0.1, -0.1) & \text{if incorrect} \end{cases}$ 
 where $\lambda = \frac{L_i - L_{budget}}{L_{budget}}$ 
 and $L_{budget} = p \cdot L_{\overline{r}} + (1 - p) \cdot L_{max}, p = \frac{c}{N}$ |
| FAST | $r_t = \begin{cases} 1 - \frac{L}{L_{avg}} & \text{if } S_{\text{difficulty}} < \theta \text{ and } r_a = 1 \\ \min(\frac{L}{L_{avg}} - 1, 1) & \text{if } \theta \leq S_{\text{difficulty}} \text{ and } r_a = 0 \\ 0 & \text{otherwise} \end{cases}$ |

# E  Human Evaluation Prompt

Image Complexity Rating Instructions for Visual Reasoning Tasks

Please rate the complexity of the given image on a scale of 1-5, considering how challenging it would be for visual reasoning tasks. Focus on aspects that affect the difficulty of analyzing, interpreting, and reasoning about the image content.

**Rating Scale:**

**1 - Very Simple**

- Clear, uncluttered images with few objects
- Simple spatial relationships
- High contrast and clear visibility
- Minimal text or numbers if present
- Straightforward visual patterns

**2 - Somewhat Simple**

- Moderately clear images with a manageable number of objects
- Basic spatial relationships requiring minimal analysis
- Good visibility with minor distractions
- Limited text or numerical information
- Recognizable patterns with minimal complexity

**3 - Moderate Complexity**

- Multiple objects with varied relationships
- Moderate spatial reasoning required
- Some visual clutter or distractions
- Moderate amount of text, numbers, or symbols
- Patterns requiring some analysis

**4 - Complex**

- Numerous objects with intricate relationships
- Challenging spatial reasoning required
- Significant visual clutter
- Substantial text, numbers, or symbols requiring careful reading
- Complex patterns requiring detailed analysis

**5 - Very Complex**

- Dense arrangement of many objects with intricate relationships
- Advanced spatial reasoning required
- Heavy visual clutter making object identification difficult
- Extensive text, numbers, or symbols with complex relationships
- Intricate patterns requiring sophisticated analysis

When rating, consider: number of objects, visual clarity, amount of information, spatial relationships, and reasoning steps needed to understand the image content.

# F  Naive GRPO Results

As shown in Figure 7, the training accuracy for naive GRPO continues to increase throughout the training process, similar to other methods like Dynamic Sampling and FAST. However, when we

Figure 7: Training accuracy of Naive GRPO.

Figure 8: Validation accuracy of Naive GRPO.

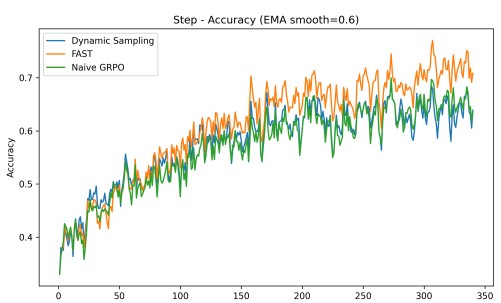

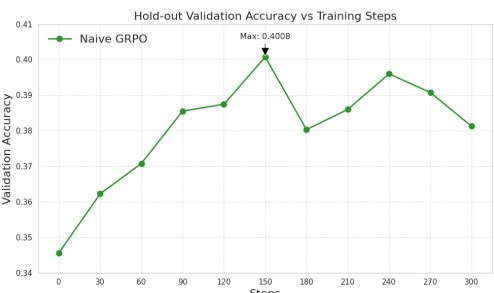

examine the validation accuracy in Figure 8, we observe a concerning trend. The validation accuracy of naive GRPO initially increases, reaching a peak of 0.40 around step 150, but then begins to decline despite continued improvements in training accuracy.

This discrepancy between training and validation performance indicates a clear **overfitting phenomenon** in naive GRPO. Upon closer examination of the model outputs in later training stages (e.g., case in Figure 9), we observe an increasing inconsistency between the chain-of-thought reasoning and the final answers produced by the model. Specifically, the model develops a tendency to generate reasoning chains that do not logically support its final conclusions, suggesting that it is memorizing training examples rather than learning generalizable reasoning patterns.

This overfitting behavior explains why the validation accuracy deteriorates even as training accuracy continues to improve. The model optimizes for the training distribution at the expense of generalization capability. Therefore, implementing early stopping based on validation performance is crucial when using naive GRPO to select the optimal checkpoint (around step 150 in our experiments) before overfitting severely impacts the model's reasoning abilities.

---

**Example Case: Inconsistent Reasoning in Naive GRPO on Training Dataset**

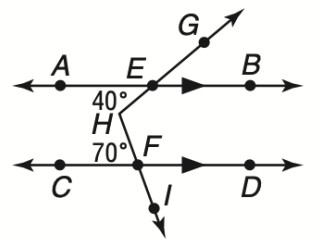

Geometry problem with angles

**Question:** What is the measure of $\angle GHI$?

**Chain of Thought:**

Since the lines AB and CD are parallel, and EF is a transversal, the angles created by the transversal with the parallel lines have specific relationships. Specifically, the angle at point E that is adjacent to 40° and the angle at point F that is adjacent to 70° are supplementary angles with the angle GHI.

The angle at point E is 40°, and since EF is a straight line, the angle adjacent to it on the other side of the transversal is 180° - 40° = 140°. Similarly, the angle at point F is 70°, and the adjacent angle on the other side of the transversal is 180° - 70° = 110°.

Since GHI is the angle formed by the intersection of the transversal with the parallel lines, it is the angle that is supplementary to the 110° angle. Therefore, the measure of angle GHI is 180° - 110° = 70°.

So, the measure of $\angle GHI$ is 70°.

**Final Answer:** 110

Figure 9: Example of reasoning inconsistency in naive GRPO on training dataset: The model produces the correct final answer (110°) despite having flawed reasoning in its CoT (which incorrectly concludes 70°). This suggests memorization rather than proper understanding of geometric principles.

# G Statistical Significance Analysis

Table 11: Main results on reasoning benchmarks. We report accuracy (Acc.) and response length (Len.) with standard deviation across 5 runs with temperature set to 0.1. * indicates statistical significance (p<0.05).

| Benchmark | Qwen2.5-VL-3B | | FAST-3B | | Qwen2.5-VL-7B | | FAST-7B | |
|---|---|---|---|---|---|---|---|---|
| | Acc. | Len. | Acc. | Len. | Acc. | Len. | Acc. | Len. |
| MathVision | 21.2 | 450.6 | 26.8±0.3* | 323.5±14.2* | 25.6 | 443.0 | 30.6±0.4* | 204.8±12.3* |
| MathVerse | 34.6 | 362.3 | 43.0±0.4* | 286.3±12.8* | 46.9 | 388.9 | 50.6±0.5* | 201.0±10.5* |
| MathVista | 62.3 | 212.9 | 66.2±0.5* | 158.7±9.3* | 68.2 | 189.1 | 73.8±0.6* | 120.7±8.2* |
| MM-Math | 33.1 | 627.9 | 39.4±0.6* | 425.0±16.7* | 34.1 | 666.7 | 44.3±0.7* | 335.6±15.3* |
| WeMath | 50.4 | 323.7 | 63.1±0.4* | 244.9±11.5* | 61.0 | 294.3 | 68.8±0.5* | 170.3±9.8* |
| DynaMath | 48.2 | 270.9 | 54.4±0.3* | 213.7±10.6* | 58.0 | 273.3 | 58.3±0.4 | 164.8±11.2* |
| MM-Vet | 61.3 | 138.8 | 64.0±0.5* | 112.7±6.9* | 67.1 | 132.5 | 71.2±0.6* | 114.1±7.5* |

To rigorously evaluate the effectiveness of our approach, we conducted statistical significance analysis across all benchmarks. Table 11 presents comprehensive results comparing our FAST models with their respective Qwen2.5-VL baselines, including standard deviations from multiple runs.

Figure 10 visualizes the performance differences between FAST-7B and Qwen2.5-VL-7B. The top panel illustrates accuracy improvements in percentage points, while the bottom panel shows response length reduction percentages. Error bars represent standard deviation across 5 runs with temperature set to 0.1, and asterisks (*) indicate statistically significant differences (p<0.05).

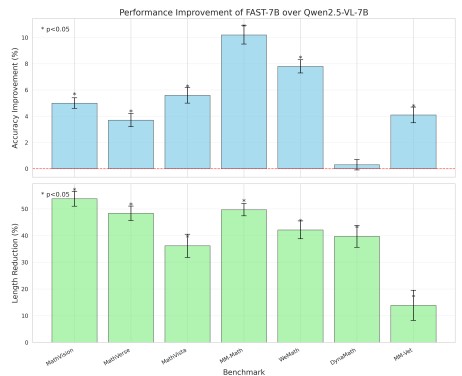

Figure 10: Performance comparison between FAST-7B and Qwen2.5-VL-7B across multiple benchmarks.

Our analysis reveals that FAST-7B achieves statistically significant accuracy improvements on 6 out of 7 benchmarks, with only DynaMath showing a non-significant improvement (0.3 percentage points). The most substantial accuracy gains are observed on mathematical reasoning tasks (MathVision: +5.0%, MathVerse: +3.7%, MM-Math: +10.2%), demonstrating our method's particular effectiveness on complex reasoning problems.

Regarding response length, FAST-7B consistently produces significantly more concise responses across all benchmarks, with length reductions ranging from 13.9% to 53.8%. This confirms that our approach successfully achieves both improved accuracy and enhanced efficiency in generating responses. The statistical significance of these improvements provides strong evidence for the effectiveness of our FAST framework in enhancing both the reasoning capabilities and efficiency.

# H Analysis of Multiplicative Difficulty Formulation and Weighted-Sum Alternative

The multiplicative combination

$$S_{\text{difficulty}} = S_{\text{extrinsic}} \cdot H_{\text{image}}$$

was designed to jointly capture a model's empirical success rate and the intrinsic visual complexity of a question. One concern raised in review is that when $S_{\text{extrinsic}}$ is high (model finds the problem hard) but $H_{\text{image}}$ is very low (visually simple), the product may be close to zero, potentially signalling "fast thinking" in a case that is actually challenging.

**Reward design avoids conflict**

As shown in Algorithm (1), the difficulty-aware length reward $r_t$ applies non-zero shaping only in two aligned cases:

- Correct and Not Complex ($S_{\text{difficulty}} < \theta$): encourage shorter responses.
- Incorrect and Complex ($S_{\text{difficulty}} \geq \theta$): encourage longer responses.

The misaligned case in question (Incorrect but Not Complex, or Correct but Complex) yields $r_t = 0$, so no penalty or incorrect encouragement is applied.

**Empirical rarity of corner cases**

We ranked $1,000$ random training samples and computed correlations between $S_{\text{extrinsic}}$, $H_{\text{image}}$, and $S_{\text{difficulty}}$. High $S_{\text{extrinsic}}$ combined with low $H_{\text{image}}$ was rare ($< 5\%$ of samples).

**Weighted-sum alternative**

We compared the multiplicative form with a weighted sum:

$$S_{\text{difficulty}}^{(\text{sum})} = \alpha S_{\text{extrinsic}} + (1 - \alpha) H_{\text{image}}, \quad \alpha = 0.5.$$

Table 12 shows near-identical results.

Table 12: Multiplicative vs weighted-sum difficulty formulation.

| Formulation | MathVision | MathVista | MathVerse | Avg. Len. |
|---|---|---|---|---|
| Multiplicative | 30.6 | 73.8 | 50.6 | 175.5 |
| Weighted Sum | 29.1 | 73.9 | 50.2 | 183.8 |

These results confirm (i) corner cases are rare in our actual training distribution, (ii) reward shaping avoids contradictory signals for such cases, and (iii) performance is robust to the choice of multiplicative vs sum combination.

# I  Continuous Slow-to-Fast Sampling

In the main text (Section 5.4), we compared our Slow-to-Fast sampling strategy against alternative approaches such as Fast-to-Slow and Dynamic Sampling [53]. Here, we further contrast a *continuous* variant of Slow-to-Fast scheduling: .

**Binary Slow-to-Fast.**  In this setting, the training curriculum makes a hard switch at the halfway point of total epochs: the first half samples only hard and medium questions, and the second half incorporates easy questions, following the procedure in Algorithm 1.

**Continuous Slow-to-Fast.**  Here, the probability of drawing an easy sample, $p_{easy}$, increases linearly with the training epoch $t$ from 0 at the start to a maximum $P_{\max}$ at the final epoch:

$$p_{easy}(t) = P_{\max} \cdot \frac{t}{T},$$

where $T$ is the total number of epochs. We set $P_{\max} = 0.4$ following initial tuning, ensuring a gradual transition from hard/medium-focus to more balanced sampling.

**Results.**  Table 13 compares the two schedules under identical training settings on MATHVISTA, MATHVISION, and MATHVERSE.

**Findings.**  Continuous scheduling provides accuracy gains (e.g., +0.6pp on MATHVISTA) and increases average output length by 26%, reducing efficiency. We hypothesize that the chosen $P_{\max}$ was insufficient to sample a large enough proportion of easy questions in later epochs, limiting potential efficiency gains. Additional tuning or adaptive $P_{\max}$ may yield more favourable trade-offs.

Table 13: Binary vs. Continuous Slow-to-Fast scheduling. Accuracy (%) / Avg. length (tokens).

| Method | MATHVISTA | MATHVISION | MATHVERSE | Avg. Len. |
|---|---|---|---|---|
| Binary | 73.8 | 30.6 | 50.6 | 175.5 |
| Continuous | 74.4 | 30.9 | 51.0 | 221.2 |

## J  Scalability to Larger Models

To evaluate the scalability of FAST-GRPO beyond mid-sized LVLMs, we train and test the framework on the 32B-parameter model `Qwen-2.5-VL-32B` using the same 18K-question training set as in our main experiments. Due to compute constraints, training was stopped after 3 epochs ($\sim$1,200 GPU hours), which likely results in a sub-optimal checkpoint. We compare FAST-32B to strong slow-thinking baselines including `Vision-R1-32B` and `MM-Eureka-32B` on six benchmarks, including MM-K12 [22], a 2,000-question scientific reasoning benchmark evenly covering math, physics, chemistry, and biology. Due to compute constraints, we stopped training after three epochs (1200 GPU hours), resulting in a likely sub-optimal checkpoint.

Table 14: Performance of FAST-GRPO on 32B models compared to baselines. Accuracy (%) / Avg length (tokens).

| Model | MATHVISION | MATHVISTA | MATHVERSE | WEMATH | MM-K12 | MM-VET | Avg Acc / Len |
|---|---|---|---|---|---|---|---|
| Qwen-2.5-VL-32B | 38.4 / 651 | 71.7 / 331 | 49.9 / 550 | 69.1 / 515 | 66.8 / 840 | 71.1 / 312 | 61.1 / 533.2 |
| Vision-R1-32B | 39.1 / 976 | 76.4 / 410 | 60.9 / 818 | 74.2 / 637 | 64.8 / 1039 | 72.2 / 384 | 64.6 / 710.6 |
| MM-Eureka-32B | 34.4 / 639 | 74.8 / 352 | 56.5 / 560 | 73.4 / 524 | 72.2 / 857 | 73.4 / 344 | 64.1 / 546.0 |
| **FAST-32B** | 37.2 / 531 | 75.4 / 268 | 57.6 / 430 | 74.4 / 420 | 68.4 / 629 | 72.6 / 254 | **64.3 / 422.1** |

**Findings.** Despite shorter training, FAST-32B matches or slightly exceeds the accuracy of stronger slow-thinking baselines while using notably fewer tokens:

- Versus `Vision-R1-32B`, average output length is reduced by $\sim 40\%$ (422.1 vs. 710.6 tokens) with comparable accuracy (64.3% vs. 64.6%).

- Versus `MM-Eureka-32B`, length is reduced by $\sim 22\%$ (422.1 vs. 546.0 tokens) while slightly improving average accuracy (64.3% vs. 64.1%).

These results indicate that FAST-GRPO scales effectively to larger LVLMs, maintaining its accuracy-efficiency trade-off. We leave exploration on ultra-large ($\geq$ 70B) LVLMs for future work.

## K  Cross-Domain Evaluation

To validate FAST-GRPO beyond math-intensive benchmarks, we conducted additional experiments on science reasoning, open-domain VQA, hallucination analysis, and low-level visual perception. These evaluations were added in response to reviewer requests for broader task coverage.

### K.1  MM-K12 Scientific Reasoning

The MM-K12 benchmark [22] consists of 2,000 multimodal reasoning questions evenly covering four domains: mathematics, physics, chemistry, and biology.

**Findings.** Compared to its base model, FAST-7B improves accuracy by +8.4pp in physics, +7.0pp in chemistry, and +8.8pp in biology, while reducing output length by $\sim 33.8\%$. Even against strong slow-thinking models such as MM-Eureka-7B, accuracy remains comparable with $\sim 30\%$ fewer tokens.

### K.2  Additional General Benchmarks

We further evaluate on four benchmarks covering open-domain VQA and visual robustness:

Table 15: Accuracy (%) and average output length (tokens) on MM-K12 across subjects.

| Model | Math | Physics | Chemistry | Biology | Avg Acc / Len |
|---|---|---|---|---|---|
| Qwen-2.5-VL-7B | 58.4 | 45.4 | 56.4 | 54.0 | 53.6 / 477.6 |
| FAST-7B | 69.0 | 53.8 | 63.4 | 62.8 | 62.2 / 371.2 |
| MM-Eureka-7B | 71.2 | 56.2 | 65.2 | 65.2 | 64.5 / 537.8 |
| OpenVLThinker-7B | 63.0 | 53.8 | 60.6 | 65.0 | 60.6 / 561.0 |
| R1-OneVision-7B | 44.8 | 33.8 | 39.8 | 40.8 | 39.8 / 817.5 |
| FAST-3B | 56.0 | 50.6 | 56.2 | 57.6 | 55.1 / 318.1 |

Table 16: Performance across diverse benchmarks.

| Model | Bingo ↑ | MMHALU ↑ | MMVP ↑ | MMEval-Pro ↑ | MM-K12 ↑ |
|---|---|---|---|---|---|
| Qwen2.5-VL-7B | 3.70 | 3.50 | 47.3 | 76.0 | 53.6 |
| FAST-7B | 3.72 | 3.40 | 47.0 | 75.0 | 62.2 |
| Vision-R1-7B | 3.62 | 3.10 | 44.0 | 72.2 | – |
| MM-Eureka-7B | 3.69 | 3.20 | 46.7 | 74.8 | 64.5 |
| OpenVLThinker-7B | 3.45 | 3.00 | 46.5 | 71.5 | 60.6 |

- **Bingo Score** [49]: Open-domain VQA benchmark for hallucination analysis.
- **MMHal** [50]: Hallucination and informativeness evaluation in open-domain VQA.
- **MMVP** [51]: Low-level visual perception probing.
- **MMEval-Pro** [52]: Calibrated multimodal benchmark spanning math, science, and general VQA.

**Findings.** FAST matches or slightly outperforms strong slow-thinking baselines in open-domain VQA and hallucination-robustness benchmarks, while producing shorter outputs. This supports the observation that adaptive reasoning length mitigates hallucination risk in multimodal reasoning [61].

## L Case Study

Figure 11 illustrates how FAST balances reasoning length and accuracy. For simple coordinate identification, R1-OneVision exhibits overthinking with 349 tokens output (highlighted in green), while FAST delivers a concise 59-token solution. For complex geometry, the base model makes a critical error in angle calculations, while R1-OneVision produces a correct but verbose 676-token solution. FAST demonstrates adaptive slow thinking with a more efficient and correct 375-token solution. validating our approach's ability to adjust reasoning depth based on question complexity.

In addition to these efficiency-focused examples, we present three representative error cases illustrating the main failure categories discussed in Section 5.5: one each for *Visual Perception Failure*, *Reasoning Error Propagation*, and *Knowledge Conflict & Gap*. These cases are shown in Figures 12–14 and provide visual, task-specific instances of how such errors manifest across different problem types.

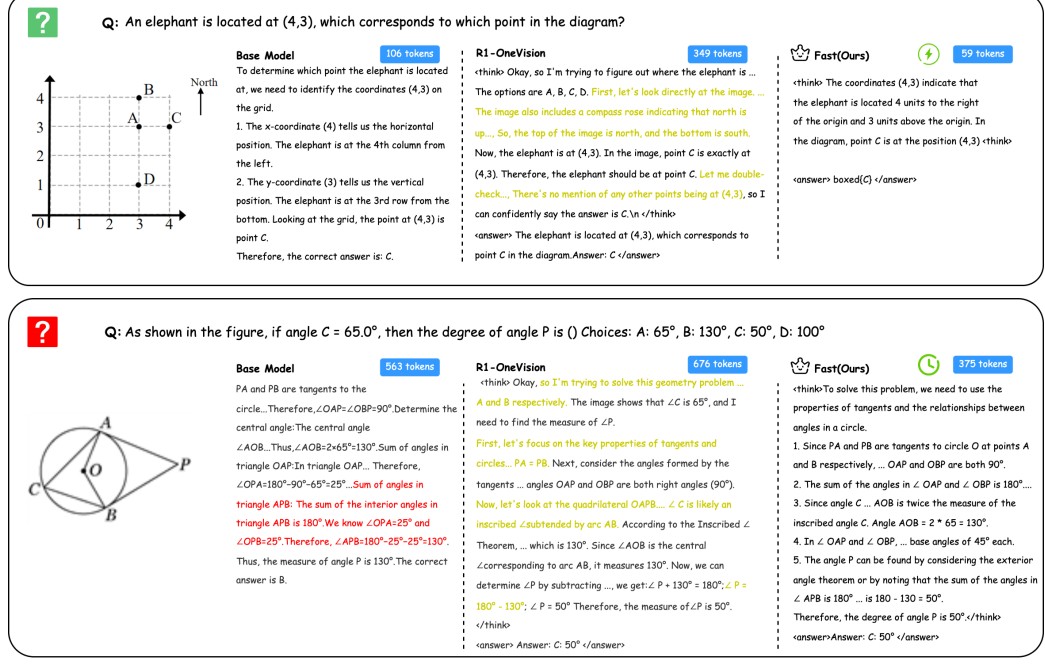

Figure 11: Case studies of different models' reasoning. Mistakes and overthinking are highlighted in red and green.

**Example Case: Visual Perception Failure**

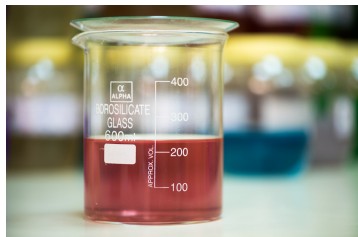

Beaker with scale markings

**Question:** *What is the highest amount this class measures?*

**Ground Truth:** 400 (ml) **Model Answer:** 600 (ml)

**Error Analysis:**

The image shows a beaker with markings up to 400 ml and a label indicating a total capacity of 600 ml. The question asks for the highest amount *measured*, which refers to the highest marked graduation (400 ml), not the container's maximum capacity. The model misinterprets the labeling and outputs 600 ml.

**Error Type:** Misinterpretation of visual context, confusing scale markings with container capacity.

Figure 12: Visual Perception Failure example: Model confuses the beaker's maximum capacity label with the highest visible measurement marking, leading to an incorrect answer. This highlights the bottleneck in visual extraction accuracy.

**Example Case: Reasoning Error Propagation**

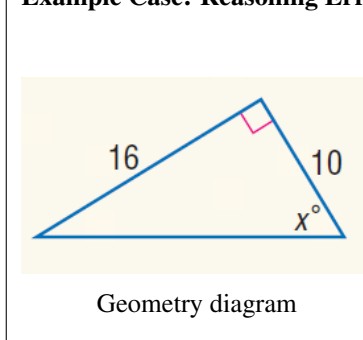

Geometry diagram

**Question:** *Find x* Choices: (A) 21, (B) 34, (C) 58, (D) 67
**Ground Truth:** (C) 58 **Model Answer:** (B) 34
**Error Analysis:**
The model correctly identifies the problem as a right-triangle angle calculation and applies the tangent ratio: $\tan x = 10/16 = 5/8$, yielding $x \approx 33.75°$. The mid-chain error occurs when mapping this computed angle to the provided options: the model selects option $34°$, overlooking that the target quantity in the diagram corresponds to another angle ($58°$). This incorrect mapping contaminates the final answer choice.
**Error Type:** A correct method with an intermediate mistake that propagates to the final conclusion.

Figure 13: Reasoning Error Propagation example: The model applies the correct trigonometric method but misaligns the computed value with the problem's actual target, causing subsequent steps to be built on a wrong assumption.

**Example Case: Knowledge Conflict & Gap**

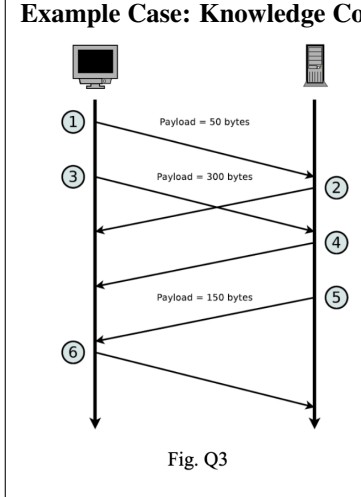

Fig. Q3

TCP transmission diagram

**Question:** *Fig.Q3 shows an excerpt of the transmission phase of a TCP connection. Assume the length of the IP header is 20 bytes. What is the ACK number at message 6?*
**Ground Truth: 839 Model Answer: 451**
**Error Analysis:**
To compute the ACK number, the model must correctly trace the TCP sequence of events in the diagram and account for byte offsets per message. Here, the model applies general TCP knowledge and standard ACK progression, but fails to interpret the specific visual data flow in the diagram (misidentifying the sequence range of message 3). This leads to an underestimated ACK number.
**Error Type:** Domain knowledge misapplication, relying on generic protocol rules over conflicting visual evidence from the figure.

Figure 14: Knowledge Conflict & Gap example: The model ignores the specific sequence number information in the visual diagram and instead applies an incorrect general TCP rule, leading to a wrong ACK number.

