# OpenReview forum: "Fast-Slow Thinking GRPO for Large Vision-Language Model Reasoning"
_NeurIPS.cc/2025/Conference — NeurIPS 2025 spotlight_

### Official Review · Reviewer_NQ6C · 2025-07-01

**Clarity:** 3
**Significance:** 3
**Originality:** 3
**Rating:** 4
**Confidence:** 2

**Summary:**

The paper introduces FAST-GRPO, a reinforcement-learning framework that explicitly balances “fast” (concise) and “slow” (detailed) reasoning in large vision-language models (LVLMs). It couples four ideas—(i) difficulty estimation, (ii) adaptive length reward, (iii) difficulty-aware KL regularisation, and (iv) a Slow-to-Fast curriculum sampler—to encourage short yet accurate answers. On seven math-heavy vision benchmarks (MathVision, MathVista, MM-Math, etc.), FAST-GRPO lifts average accuracy by 10 + pp while cutting answer length by 32.7 – 67.3 %. Remarkably, it reaches or surpasses strong baselines with only 18 k RL samples and no SFT fine-tuning.

**Questions:**

Further validating the framework’s scalability to larger models, improving the reliability of the difficulty estimation, and adding cross-task results would greatly enhance the paper’s credibility and practical value.

**Ethical Concerns:**

["NO or VERY MINOR ethics concerns only"]

**Final Justification:**

Thanks for the detailed rebuttal and additional experimental detail. All my concerns have been resolved.

**Limitations:**

see above.

**Quality:**

3

**Strengths And Weaknesses:**

Strengths
1、The authors are first to combine difficulty-aware length rewards, a dynamic KL schedule, and a Slow-to-Fast sampler within GRPO for LVLMs, giving explicit control over “fast vs. slow” reasoning.
2、On seven public benchmarks, accuracy rises by double digits while answers shorten substantially; ablation studies show each component is necessary.

Weakness:
1、Experiments stop at 7 B parameters; scalability to ≥ 70 B remains untested (noted in Limitations).
2、Evaluation focuses on math/diagram reasoning; generalisation to open-domain VQA, captioning, or dialogue is uncertain.

---

> ### Author Rebuttal · Authors · 2025-07-31
>
> We’d like to thank the reviewer for acknowledging the novelty and effectiveness of our proposed approach. We address your raised points as follows.
>
> > **W1,Q1: Experiments stop at 7B parameters, scalability on larger models remains untested.**
>
> We agree with your point that validating the scalability of our approach on larger models is important.
>
> While we noted the computational constraints in our original manuscript's limitations section, to directly address your concern, we **train FAST-GRPO on Qwen-2.5-VL-32B with the same training dataset. Due to time and compute constraints, we stopped training after three epochs (1200 GPU hours), resulting in a likely sub-optimal checkpoint.** We compare our method against several strong baselines on 6 benchmarks including the newly introduced MM K12 benchmark proposed in MM-Eureka [1], a comprehensive reasoning benchmark of 2,000 problems spanning math, physics, chemistry, and biology. The results are presented below:
>
> | Model           | MathVision (Acc/Len) | MathVista (Acc/Len) | MathVerse (Acc/Len) | WeMath (Acc/Len) | MM-K12 (Acc/Len) | MM-Vet (Acc/Len) | Avg Acc | Avg Len |
> |-----------------|----------------------|----------------------|----------------------|------------------|------------------|------------------|---------|---------|
> | Qwen-2.5-VL-32B | 38.4 / 651           | 71.7 / 331           | 49.9 / 550           | 69.1 / 515       | 66.8 / 840     | 71.1 / 312       | 61.1    | 533.2   |
> | Vision-R1-32B   | 39.1 / 976           | 76.4 / 410           | 60.9 / 818           | 74.2 / 637       | 64.8 / 1039      | 72.2 / 384       | 64.6    | 710.6   |
> | MM-Eureka-32B   | 34.4 / 639           | 74.8 / 352           | 56.5 / 560           | 73.4 / 524       | 72.2 / 857     | 73.4 / 344       | 64.1    | 546.0   |
> | **FAST-32B**    | 37.2 / 531           | 75.4 / 268           | 57.6 / 430           | 74.4 / 420       | 68.4 / 629    | 72.6 / 254       | **64.3**    | **422.1**   |
>
> These new experimental results demonstrate the effectiveness and scalability of our approach:
>
> 1.  **Comparable or Superior Reasoning Accuracy:** Compared to strong "slow-thinking" models like Vision-R1-32B and MM-Eureka-32B, FAST-32B achieves comparable or even better average accuracy across various reasoning and VQA benchmarks.
> 2.  **Significant Efficiency Gains:** The average response length of FAST-32B (422.1 tokens) is nearly 40% shorter than that of Vision-R1-32B (710.6 tokens) and 22% shorter than that of MM-Eureka-32B (546.0 tokens), achieving a balance for both performance and efficiency.
>
> -----
>
> > **Q2: Improving the reliability of the difficulty estimation.**
>
> Thank you for your question regarding the reliability of our difficulty estimation. Our difficulty estimation framework is composed of two complementary components to ensure its reliability:
>
> 1.  **Extrinsic Difficulty:** This metric is derived from the model's performance (*pass@k* on a given problem). As this metric is derived directly from the model's actual performance, it accurately reflects the model's empirical problem-solving ability.
> 2.  **Intrinsic Difficulty:** This metric captures the inherent visual complexity of an image using GLCM and ViT entropy. The reliability of our intrinsic metric is validated from two perspectives:
>       * The GLCM component is established to have strong human alignment ($SRCC=0.75, PLCC=0.77$)[2].
>       * As detailed in Section 5.4 Validation of Image Complexity, our human annotation study confirmed that our combined image complexity metric has a moderate and statistically significant correlation with human judgments ($SRCC=0.49, PLCC=0.50$).
>
> By combining these two metrics, our approach provides a comprehensive and reliable estimation of problem difficulty.
>
> -----
>
> > **W2,Q3: Evaluation focues on math/diagram reasoning; Generalisation to open-domain VQA, captioning or dialogue is uncertain.**
>
> While our initial focus was on enhancing visual reasoning across math, science, VQA, and figure understanding, we agree that demonstrating broader generalization capability strengthens the value of our work.
>
> First, we would like to clarify that, as GRPO typically requires verifiable reward signals, our training dataset did not include open-domain VQA samples. Despite this, our method was evaluated on the MM-Vet benchmark, an open-domain VQA benchmark, where FAST-7B shows gains over the base model (71.2% vs. 67.1%).
>
> Second, to address your concerns, we have conducted a comprehensive set of **experiments on 5 additional benchmarks**, which cover diverse tasks including open-domain VQA, hallucination analysis, low-level visual perception, and scientific reasoning. The additional benchmarks are:
>
>   * **Bingo Score [3]:** An open-domain VQA benchmark designed for holistic hallucination analysis.
>   * **MMHal Benchmark [4]:** An open domain VQA benchmark, evaluating both hallucination and informativeness to calculate an overall socre.
>   * **MMVP [5]:** A diagnostic benchmark that evaluates low-level visual perception capability of LVLMs.
>   * **MMEval-Pro [6]:** A comprehensive calibrated evaluation benchmark, adapted from sources including MathVista, MMMU, ScienceQA and manually crafted questions.
>   * **MM K12 [1]:** A Human-annotated multimodal scientific reasoning benchmark across math, physics, chemistry and biology.
>
> We compare FAST-3B and FAST-7B against strong slow-thinking baselines, with the results below:
>
> | Model            | Bingo ↑ | MMHALU ↑ | MMVP ↑ | MMEval-Pro ↑ | MM K12 ↑ |
> |------------------|:-------:|:--------:|:------:|:------------:|:--------:|
> | Qwen2.5-VL-3B    |  3.64   |  3.33    |  36.0  |   67.8       |   —      |
> | LLM-R1-3B        |  3.58   |  2.78    |  28.7  |   66.9       |   —      |
> | Curr-REFT-3B     |  3.52   |  2.27    |  24.0  |   60.2       |   —      |
> | **FAST-3B**      |  **3.60**   |  **3.10**    |  **37.2**  |   **67.1**       |  **55.1**    |
> | Qwen2.5-VL-7B    |  3.70   |  3.50    |  47.3  |   76.0       |  53.6    |
> | Vision-R1-7B     |  3.62   |  3.10    |  44.0  |   72.2       |   —      |
> | R1-OneVision-7B  |  3.65   |  3.20    |  43.0  |   69.4       |  39.8    |
> | OpenVLThinker-7B |  3.45   |  3.00    |  46.5  |   71.5       |  60.6    |
> | MM-Eureka-7B     |  3.69   |  3.20    |  46.7  |   74.8       |  64.5    |
> | **FAST-7B**      |  **3.72**   |  **3.40**  |  **47.0**  |   **75.0**       |  **62.2**    |
>
> These results on new benchmarks demonstrate that:
>
> 1.  **Our method has strong generalization capabilities in scientific reasoning.** The performance of FAST on MM K12, which covers physics, chemistry, and biology, shows that our method generalizes well beyond math to other scientific domains, outperforming strong baselines.
> 2.  **Our method outperforms slow-thinking LVLMs in open-domain VQA tasks.** On the hallucination benchmarks (Bingo Score and MMHALU), visual perception, and general VQA and benchmarks (MMVP and MMEval-Pro), slow-thinking LVLMs tend to significantly degrade, while our FAST models achieve higher scores. **This observation corroborates the findings of Liu et al. [7], who noted that lengthy reasoning chains in slow-thinking models can amplify visual hallucinations**.
>
>
> [1] Meng, Fanqing, et al. "Mm-eureka: Exploring the frontiers of multimodal reasoning with rule-based reinforcement learning." arXiv:2503.
>
> [2] Zhang, Jinjin, et al. "Diffusion-4k: Ultra-high-resolution image synthesis with latent diffusion models." CVPR. 2025.
>
> [3] Cui, Chenhang, et al. "Holistic analysis of hallucination in gpt-4v (ision): Bias and interference challenges." arXiv preprint arXiv:2311.03287 (2023).
>
> [4] Sun, Zhiqing, et al. "Aligning Large Multimodal Models with Factually Augmented RLHF." ACL. 2024.
>
> [5] Tong, Shengbang, et al. "Eyes wide shut? exploring the visual shortcomings of multimodal llms." CVPR. 2024.
>
> [6] Huang, Jinsheng, et al. "MMEVALPRO: Calibrating Multimodal Benchmarks Towards Trustworthy and Efficient Evaluation." NAACL, 2025
>
> [7] Liu, Chengzhi, et al. "More Thinking, Less Seeing? Assessing Amplified Hallucination in Multimodal Reasoning Models." arXiv:2505.

---

> ### Author Response · Authors · 2025-08-06
>
> Dear Reviewer,
>
> We hope this message finds you well. **As the discussion period is nearing its end**, we wanted to ensure we have addressed all your concerns satisfactorily. If there are any additional points or feedback you'd like us to consider, please let us know. Your insights are invaluable to us, and we're eager to address any remaining issues to improve our work.
>
> Thank you for your time and effort in reviewing our paper.

---

> ### Author Response · Authors · 2025-08-07
>
> Dear Reviewer NQ6C,
>
> Thank you once again for your valuable comments on our submission. **As the discussion phase is approaching its end, we would like to kindly confirm whether we have sufficiently addressed all of your concerns**. Should there be any remaining questions or areas requiring further clarification, please do not hesitate to let us know. If you are satisfied with our responses, we would greatly appreciate your consideration in adjusting the evaluation scores accordingly.
>
> We sincerely look forward to your feedback.

---

> ### Comment · Area_Chair_uc12 · 2025-08-08
> **AC Message**
>
> Dear Reviewer,
>
> The authors have already submitted their response to your review, and we are approaching the deadline for author-reviewer discussions (August 8, 11:59 PM AoE).
>
> Could you please share your feedback with the authors regarding whether their response sufficiently addresses the concerns raised in your review, and if any issues remain that require further clarification or elaboration?
>
> As a key reminder, active engagement in the discussion process is critical at this stage. Reviewers must participate in discussions with authors (e.g., by posting follow-up feedback in the discussion thread), and then submit a Mandatory Acknowledgement to confirm your participation.
>
> Best regards,
>
> AC

---

### Official Review · Reviewer_jkkr · 2025-07-02

**Clarity:** 3
**Significance:** 3
**Originality:** 3
**Rating:** 5
**Confidence:** 4

**Summary:**

1. The authors identify a key problem in current Large Vision-Language Models (LVLMs): "slow-thinking" models often "overthink" by generating overly verbose responses for all questions, regardless of their actual difficulty. This can be inefficient and even hurt accuracy on simpler problems. To address this, the authors propose FAST-GRPO- a method that enables the model to adapt its reasoning depth. The core of the approach is to first use two metrics to estimate the difficulty of a question. This difficulty estimation then guides GRPO process in two ways: adaptive length-based rewards to encourage concise answers for easy questions and more detailed reasoning for difficult ones. It uses a difficulty-aware KL divergence to control how much the model's policy can change, allowing for more exploration on complex problems.
2. The proposed approach forces the model to learn when to apply "fast thinking" (short answers) and when to use "slow thinking" (detailed reasoning), leading to higher accuracy while significantly reducing the average response length.
3. Two difficulty metrics are introduced - (a) Extrinsic Difficulty: Based on model's pass@k rate (how often it answers correctly), (b) Intrinsic Difficulty: Based on image complexity, using GLCM entropy and ViT entropy.
4. A "Slow-to-Fast" curriculum is used—early training focuses on hard questions to build depth, then shifts to easier ones to promote concise thinking.
5. The reward function combines - Accuracy reward, Format reward and Length reward: Encourages brevity on easy questions and thoroughness on hard ones. he KL coefficient is dynamically adjusted—looser for hard questions to allow exploration, tighter for easy ones to prevent overthinking.
The model is trained on 18K multimodal reasoning questions (filtered and difficulty-stratified), and evaluated on 7 benchmarks including MathVista, MathVerse, and MM-Vet.

**Questions:**

1. Could you provide a more robust justification for the multiplicative formulation of the complexity score (S_complexity)? Have you experimented with alternative formulations, such as a weighted sum?
2. The "Slow-to-Fast" curriculum is a binary split. Have you considered a more continuous curriculum where the probability of sampling "Easy" questions gradually increases over the course of training?
3. The ablation study (Table 5) shows that removing Data Sampling causes the largest performance drop. Given the aggressiveness of the filtering, to what extent are the final gains attributable to the highly curated nature of the 18K dataset versus the FAST-GRPO algorithm itself?

**Ethical Concerns:**

["NO or VERY MINOR ethics concerns only"]

**Final Justification:**

Some of the questions related to boundary case in reward and continuous reward are addressed with further experimental details. Further, during the discussion period, the authors have shared further error analysis which very critical and useful to understand the improvement with FAST-7B in reasoning over easy, medium, hard while showing the primary gap in perception errors of these models. Given this further analysis, I'm modifying my rating.

**Limitations:**

Yes

**Paper Formatting Concerns:**

The Algorithm 1 should be updated to indicate when does fast-thinking start (boundaries are not defined in the algorithm)

**Quality:**

3

**Strengths And Weaknesses:**

Strengths-
1. Introduces a dynamic RL-based method (FAST-GRPO) that intelligently adapts reasoning depth based on question difficulty—filling a key gap in LVLM reasoning literature.
2. The paper proposes innovative, complementary difficulty metrics (extrinsic and intrinsic), including a practical image complexity estimation using GLCM and ViT entropy.
3. The empirical results show that on multiple LVLM reasoning benchmarks, the proposed approach provides SOTA accuracy while significantly reducing token usage compared to existing slow-thinking methods.
4. The insights presented in the prelimnary investigation are useful.

Areas of Improvement-
1. The paper's method for calculating complexity, using the formula $S_{complexity} = S_{difficulty} \cdot H_{img}$, contains a potential flaw in its logic. The issue arises when a problem is very difficult for the model to solve (resulting in a high $S_{difficulty}$) but is paired with a visually simple image (leading to a very low $H_{img}$). Because of the multiplication, the final complexity score for this problem would be close to zero. According to the framework's design, a low complexity score signals the model to use "fast thinking" and rewards shorter, more concise answers. This creates a direct contradiction: the model is encouraged to provide a brief solution for a problem that it finds very hard and likely needs a more detailed, step-by-step reasoning process ("slow thinking") to solve correctly. This conflicting reward signal could hinder the model's ability to learn how to tackle such challenging problems effectively.
2. The authors start with ~500K questions and, after two filtering stages, arrive at a final training set of just 18K questions. However, the exact details of filtering strategy is not laid out well.
3. The current curriculum learning approach from slow-fast has a binary threshold in terms of epochs, however, the authors do not experiment with a graded approach of switching from slow to fast thinking.
4. All experiments are conducted on models up to 7B parameters; it’s unclear whether FAST-GRPO would maintain gains on larger-scale LVLMs (e.g., 70B+).
5. The paper lacks sufficient qualitative analysis and error breakdown. While it reports strong quantitative improvements, it provides only a single example of reasoning inconsistency and does not include comparative qualitative outputs that illustrate how FAST-GRPO adapts reasoning depth in practice. Additionally, there is no systematic error analysis to identify failure modes across question types or reasoning stages, which limits interpretability and diagnostic insights into model behavior.

---

> ### Author Rebuttal · Authors · 2025-07-31
>
> We’d like to thank the reviewer for acknowledging the novelty and effectiveness of our proposed approach. We address your raised insightful points as follows.
>
> > **W1, Q1: The potential conflicting reward signal could hinder the model's ability to learn such challenging problems. Have you experimented with alternative formulations, such as a weighted sum?**
>
> To ensure clarity, we will adopt the reviewer's terminology for this response, where $S_{difficulty}$ refers to the model's extrinsic difficulty (termed $S_{extrinsic}$ in our paper) and $S_{complexity}$ refers to the final combined score (termed $S_{difficulty}$ in our paper).
>
> In our method design, the corner case you proposed, a problem with high difficulty ($S_{difficulty}$) but low visual complexity ($H_{image}$) will **indeed result in a low final complexity score ($S_{complexity}$). However, we will demonstrate that this does not create a conflicting reward signal.**
>
> Our analysis is twofold:
> 1. Our **reward design** avoids conflicting signals from this corner case.
> 2. **Empirical evidence** shows that using a weighted sum yields nearly identical results, suggesting the practical impact is negligible.
> ---
>
> **Analysis 1: Reward Design Avoids Conflicting Signals**
>
> As detailed in the below reward equation (Equation 4 in paper), our difficulty-aware length reward ($r_t$) is applied based on three distinct cases:
>
> $$r_t =
> \begin{cases}
> 1 - \frac{L}{L_{avg}} & \text{if } (S_d < \theta) \wedge (r_a = 1) \\\\
> \min\left(\frac{L}{L_{avg}} - 1, 1\right) & \text{if } (S_d \ge \theta) \wedge (r_a = 0) \\\\
> 0 & \text{otherwise}
> \end{cases}$$
>
> Case 1:  When the solution is **correct** and the question is categorized as **"Not Complex"** ($S_{complexity} < \theta$), the length reward $r_t$ encourages shorter responses. Specifically, it penalizes responses where the length $L$ exceeds the batch average length $L_{avg}$.
>
> Case 2:  When the solution is **incorrect** and the question is categorized as **"Complex"** ($S_{complexity} \ge \theta$), the reward $r_t$ moderately encourages longer responses.
>
> Case 3:  Otherwise. This category includes two scenarios: **a) the solution is incorrect and the question is "Not Complex"**, and b) the solution is correct and the question is "Complex". In these cases, $r_t$ is 0, providing neither a reward nor a penalty.
>
> First, since the problem is inherently difficult for the model, the majority of its sampled responses will be incorrect ($r_a=0$). This situation falls into **Case 3**. In this case, the length reward $r_t$ is zero. This means the model is **not penalized for generating longer, exploratory reasoning chains for most of its attempts**. This directly avoids the conflicting reward you were concerned about.
>
> Second, in the rarer event that the model produces a correct solution ($r_a=1$), it falls into **Case 1**. Here, a modest length penalty is only applied if the response is longer than the batch average length $L_{avg}$. Additionally, **the final reward is dominated by the primary accuracy reward ($r_a=1$)**. This is because, as defined in our reward ($r = r_a + \lambda_r \cdot r_f + \lambda_t \cdot r_t$), other rewards (format and length) are scaled by coefficients of 0.5.
>
> ---
>
> **Analysis 2: Empirical Evidence Shows This Corner Case Has Negligible Impact**
>
> To validate our design, we ranked 1,000 random training samples to analyze the correlation between different scoring metrics. We focused on two key aspects: **(1)** the ranking correlation between $S_{difficulty}$ and $H_{image}$ to evaluate potential corner cases, and **(2)** the ranking correlation between multiplicative and the weighted sum. Our correlation metrics included Spearman's Rank Correlation Coefficient (SRCC), Pearson's Linear Correlation Coefficient (PLCC), and Intersection@top-200.
>
> The results provide insights about our data distribution.
> * **First**, there is a strong positive correlation between $S_{difficulty}$ and $H_{image}$, which suggests the reviewer's proposed corner case is infrequent in our data.
> * **Second**, the two formulations (multiplicative vs. weighted sum) have a high correlation and identify 193 of the same top 200 most complex questions.
>
> | |SRCC| PLCC | Intersect@top 200 |
> |:----:|:-:| :-: | :-: |
> | **$S_{difficulty}$ vs. $H_{image}$** | 0.75 | 0.68 | 156 |
> | **Multiplicative vs. Weighted sum** | 0.94 | 0.92 | 193 |
>
> Finally, we **experimented FAST-GRPO with the weighted-sum formulation ($S_{complexity}=S_{difficulty}+ H_{image}$)**. As shown below, a specific formulation has no significant impact on the model's performance.
>
> | | MathVision | MathVista | MathVerse | Avg. Len. |
> | :--- | :---: | :---: | :---: | :---: |
> | **Multiplicative** | 30.6 | 73.8 | 50.6 | 175.5 |
> | **Weighted Sum** | 29.1 | 73.9 | 50.2 | 183.8 |
>
> In summary, **our analysis shows that while the corner case is theoretically possible, it is rare in our data distribution, and our method's performance is robust to the specific formulations.** We will incorporate these results and analysis into the revision.
>
> ---
> > **W2: Filtering strategy is not laid out well.**
>
> To filter our initial 500K-question dataset, we applied a rigorous filtering process. First, we deduplicated all samples with duplicate images and removed any questions that overlapped with our evaluation benchmarks. Next, to satisfy the GRPO algorithm's need for verifiable rewards, we standardized the data to include only closed-form questions—such as multiple-choice, numerical, or single-word answers—using regular expressions and custom rules. Finally, we filtered by extrinsic difficulty. We measured our base model's success rate for each question (pass@8) and removed those that were either too easy (pass@8 = 1) or too hard (pass@8 = 0), as these provide a poor learning signal for training. Due to limited space, further specifics will be in the revision.
>
> ---
> > **W3, Q2: A graded approach to switching from slow to fast thinking.**
>
> In response to this feedback, we have designed and conducted experiments to compare our original binary strategy with a **continuous Slow-to-Fast** curriculum. In this new setup, we introduce a linear scheduling for sampling "Easy" questions. The probability of sampling an easy question, $P_{easy}$, increases linearly with the training epoch $e$ (from $0$ to total epoches $E=10$), following $P_{easy}(e) = P_{max} \cdot (e/E)$, where $P_{max}$ is a hyperparameter denotes the max sampling probability of easy questions, set to 0.4. This ensures a gradual transition from focusing on hard and medium samples to incorporating more easy samples as training progresses.
>
> We evaluated this continuous approach on three benchmarks, with the other settings identical to our main experiments. The results are presented below:
>
> |  | MathVista | MathVision | MathVerse | Avg. Length |
> | :--- | :---: | :---: | :---: | :---: |
> | **Binary** | 73.8 | 30.6 | 50.6 | 175.5 |
> | **Continuous** | 74.4 | 30.9 | 51.0 | 221.2 |
>
> As shown in the table, the continuous curriculum can **surpass the binary method in accuracy, while with longer responses.** We hypothesize this is because the  $P_{max}$ is not sufficiently high to sample adequate easy questions. We will incorporate results into the revision.
>
> ---
> > **W4: FAST-GRPO on Larger-scale model.**
>
> **Thank you for the question. As this point is similar to the one raised in comment W1,Q1 by Reviewer NQ6C, we respectfully refer you to our detailed response provided there.**
>
> ---
> > **W5: Lack qualitative analysis.**
>
> We'd like to respectfully point out that we did provide qualitative analyses in the appendix, which are:
>
> * **Case Study of Fast-Slow Thinking (Appendix H, Figure 10):** This section provides a direct, side-by-side comparison of our FAST model against the Base Model and a "slow-thinking" model (R1-OneVision) on two distinct problems.
>     * On a simple question (coordinate identification), our model provides a correct and concise 59-token answer, while R1-OneVision exhibits overthinking with a needlessly verbose 349-token response.
>     * On a relatively complex geometry problem, the Base Model fails, and while R1-OneVision is correct, it is again verbose (676 tokens). In contrast, our FAST model adaptively adopts slow thinking, producing a correct and significantly more efficient 375-token solution.
>
> * **Analysis of Naive GRPO Failure (Appendix F, Figure 8):** We show a case where naive GRPO produces a correct final answer but with a logically flawed chain-of-thought, which highlights the overfitting problem that FAST-GRPO can effectively alleviate.
>
> **Due to the rebuttal's restrictions (e.g., no figures), we will provide more qualitative analysis, including case study and failure pattern in the revised version.**
>
> ---
> > **Q3: Final gains from dataset curation or FAST-GRPO algorithm.**
>
> First, to clarify the point raised from **Table 5 in Section 5.3 Ablations**, the term **Data Sampling** refers specifically to the Slow-to-Fast sampling strategy used during training. This is **distinct from the one-time data filtering** we performed as a preprocessing step to create the 18K training set used across all relevant experiments.
>
> Second, to precisely isolate and quantify the contribution of our algorithm, we designed the **Naive GRPO** baseline. Specifically, this baseline is trained on the **same 18K curated dataset**, without the core components of our method.
>
> The results from Table 5 demonstrate a significant performance uplift: Compared to the naive baseline, FAST-GRPO simultaneously boosts accuracy by +3.4 (MathVista), +2.5 (MathVision) points, while also reducing the response length from 243.6 to 175.5 tokens.
>
> In conclusion, while a well-curated dataset provides a strong foundation, **the FAST-GRPO algorithm itself mainly contributes to improvement in reasoning accuracy and efficiency**.
>
> ---
> > **Format Concern: boundaries are not defined in the algorithm**
>
> The boundary is half of the total epochs. We will revise it in the revision.

---

> ### Author Response · Authors · 2025-08-06
>
> Dear Reviewer,
>
> We hope this message finds you well. **As the discussion period is nearing its end**, we wanted to ensure we have addressed all your concerns satisfactorily. If there are any additional points or feedback you'd like us to consider, please let us know. Your insights are invaluable to us, and we're eager to address any remaining issues to improve our work.
>
> Thank you for your time and effort in reviewing our paper.

---

> > ### Comment · Reviewer_jkkr · 2025-08-06
> >
> > Thank you for the detailed rebuttal and additional experimental details. I appreciate the thoughtful responses to the concerns raised. Some of the questions related to boundary case in reward and continuous reward are addressed with further experimental details. However, the points related to lack on qualitative insights (not only the couple of case-studies shared in Appendix) still remain open. I will take these into consideration in shaping my final rating and justification.

---

> > > ### Author Response · Authors · 2025-08-07
> > > **More Qualitative Analysis and Error Breakdown (2/2)**
> > >
> > > ### **More Qualitative Analysis and Error Breakdown (2/2)**
> > >
> > > We then **briefly demonstrates FAST-GRPO successfully implements adaptive reasoning**.
> > >
> > > | Test | Qwen2.5-VL (Acc.) | Qwen2.5-VL (Len.) | R1-OneVision (Acc.) | R1-OneVision (Len.) | FAST (Acc.) | FAST (Len.) |
> > > |:----:|:-----------------:|:-----------------:|:-------------------:|:-------------------:|:-----------:|:-----------:|
> > > | Easy | 72.7              | 318               | 69.5                | 623                 | 78.2        | 189         |
> > > | Med  | 33.9              | 406               | 40.4                | 661                 | 49.2        | 220         |
> > > | Hard | 5.5               | 412               | 10.2                | 835                 | 12.3        | 304         |
> > > | All  | 37.7              | 378               | 40.3                | 731                 | 46.4        | 239         |
> > >
> > >
> > >
> > > First, table 1 in paper highlights our model's **key qualitative advantage: its adaptive 'fast-slow thinking'**. It defaults to efficient 'fast thinking' for simple tasks (189 tokens) but strategically extends its reasoning for complex ones as a form of 'slow thinking' (304 tokens). This dynamic shift is the core reason it achieves strong accuracy with enhanced efficiency.
> > >
> > >
> > >
> > > | Model | Geometry Reasoning | Algebraic Reasoning | Geometry Problem Solving | Math Word Problems | Overall |
> > > |---|---|---|---|---|---|
> > > | Qwen2.5-VL-7B | 66.9 | 68.7  | 66.8  |76.9|68.2|
> > > | FAST-7B | 78.8 | 78.2 | 78.4 | 80.1 |73.8|
> > >
> > >
> > > Second, we further compare the **improvements of FAST-7B on sub-tasks of MathVista**. A key qualitative insight is that FAST-7B achieves its **largest performance gains in the most abstract reasoning categories**, even while reducing its average response length. As shown in the table, it boosts accuracy on challenging sub-tasks like Geometry Reasoning (from 66.9 to 78.8) and Algebraic Reasoning (from 68.7 to 78.2).

---

> > > ### Author Response · Authors · 2025-08-07
> > >
> > > Once again, we extend our heartfelt thanks for your valuable time and constructive feedback on our work. Your feedback is of great significance to us, and we are happy to continue our discussions to jointly refine our research.

---

> > > ### Author Response · Authors · 2025-08-09
> > > **Follow-up on Quality Analysis for Our Submission**
> > >
> > > Thank you for your detailed feedback and constructive comments on our submission. To address your concern on qualitative insights, in response, we have conducted an additional quality analysis.
> > >
> > > The new results are included in the discussion thread. If you find that this additional analysis resolves your concern, we would greatly appreciate it if you could consider updating your final rating accordingly.
> > >
> > > Thank you again for your time, effort, and valuable input during the review process.
> > >
> > > Best regards,
> > >
> > > The authors of Submission 14785

---

> ### Author Response · Authors · 2025-08-07
> **More Qualitative Analysis and Error Breakdown (1/2)**
>
> Thank you for your valuable and constructive feedback.
>
> ### **More Qualitative Analysis and Error Breakdown (1/2)**
>
> Beyond the case studies in appendix, we further qualitatively analyse the model's bahavior. We **first present a systematic breakdown of the model's failure modes**.
>
> Our error breakdown of **failure cases of FAST-7B, R1-OneVision-7B and base model (Qwen2.5-VL-7B) on MathVista** identifies three predominant categories:
>
> * **Visual Perception Failure:** Models incorrectly extract or interpret visual information.
> * **Reasoning Error Propagation:** Models often can't backtrack to fix a mistake made midway through their reasoning.
> * **Knowledge Confliction & Knowledge Gap:** Models either **prioritises internal knowledge over conflicting visual evidence** or **hallucinate a response when faced with a knowledge gap**.
>
> | Model | Visual Perception Failure | Reasoning Error Propagation | Knowledge Confliction & Knowledge Gap | All |
> | :--- | :--- | :--- | :--- | :--- |
> | FAST | 141 | 74 | 57 | 272 |
> | R1-OneVision | 177 | 102 | 80 | 359 |
> | Base | 139 | 105 | 74 | 318 |
>
>
> **Insight Analysis**
>
> * **Reduction in Reasoning & Knowledge Errors**: Compared to the baseline, FAST-7B reduces failures from `Reasoning Error Propagation` and `Knowledge Confliction` by ~27% and ~19%, respectively. This is attributed to its **adaptive thinking**: shorter logical chains minimize opportunities for error propagation and effectively suppress hallucinations caused by long reasoning chains that contradict visual evidence.
>
> * **Visual Perception as the Primary Bottleneck**: Over 51% of all failures in FAST-7B are due to `Visual Perception Failure`. This demonstrates that if the model initially "sees" the context incorrectly, even a flawless reasoning process will lead to a wrong answer. Therefore, enhancing fundamental visual capability is a crucial factor for future improvement.
>
>
> ***
>
> We provide specific cases for each failure mode below.
>
> ### **Failure Mode 1: Visual Perception Failure**
>
> * **Case A: Misinterpreting Visual Context**
>     * **Question**: "What is the highest amount this class measures?"
>     * **Image**: A beaker with measurement lines up to 400ml and a label indicating a 600ml total capacity.
>     * **Error**: The model sees both "400ml" and "600ml" but **fails to distinguish between a marked measurement and the container's total capacity**, incorrectly answering 600.
>
> * **Case B: Incorrect Data Extraction from Charts**
>     * **Question**: "How many bars have value below 40?"
>     * **Image**: A bar chart.
>     * **Error**: The model correctly identifies three bars below the threshold but **erroneously includes a fourth bar (value 42.1) in its count**, resulting in a wrong total.
>
> ---
>
> ### **Failure Mode 2: Reasoning Error Propagation**
>
> * **Case A: Mid-Reasoning Comparison Error**
>     * **Question**: "Is the sum of the smallest two bars greater than the largest bar?"
>     * **Image**: A bar chart.
>     * **Error**: The model correctly identifies the values and calculates their sum (9.29% + 12.51% = 21.8%). However, **it fails at the final logical comparison** (comparing 21.8% to the largest bar, 21.37%), leading to the wrong 'Yes'/'No' answer.
>
> * **Case B: Hallucinating a Geometric Rule**
>     * **Question**: A geometry problem asking to find an angle.
>     * **Image**: A diagram with parallel lines and a triangle.
>     * **Error**: The model **abandons a valid reasoning path to invent a false geometric rule**. It then performs a correct calculation based on this flawed premise and fails to self-correct, producing an incorrect answer.
>
> ---
>
> ### **Failure Mode 3: Knowledge Confliction & Knowledge Gap**
>
>
> * **Case A: Confident Assertion of Incorrect Knowledge**
>     * **Question**: "In which part of the mold are the cylindrical ports located?"
>     * **Image**: A diagram clearly showing the ports in the lower half.
>     * **Error**: The model **ignores the direct visual evidence** and instead asserts its incorrect "general knowledge" from its language model that ports are "typically located in the upper part," confidently selecting the wrong answer.
>
> * **Case B: Flawed Application of Domain-Specific Knowledge**
>     * **Question**: "What is the ACK number at message 6?"
>     * **Image**: A TCP transmission sequence diagram.
>     * **Error**: The model understands the TCP protocol in theory but **fails to apply that knowledge correctly to the specific data flow in the diagram**. It performs a flawed calculation using the wrong values.

---

### Official Review · Reviewer_A2xT · 2025-07-03

**Clarity:** 3
**Significance:** 3
**Originality:** 3
**Rating:** 5
**Confidence:** 4

**Summary:**

* This paper proposes FAST-GRPO, which is a variant of the GRPO reinforcement learning method that balances fast and slow reasoning for LVLMs through incorporating adaptive length-based rewards and dynamic regularization determined by question characteristics.
* To determine the difficulty of questions, they propose 2 metrics — extrinsic and intrinsic difficulty — which provides accurate feedback to guide the model to choose between fast and slow thinking.
* They improve the GRPO algorithm through including adaptive length-based rewards and difficulty-aware KL divergence.
  * Empirical results on 7 reasoning benchmarks show the effectiveness of the proposed FAST-GRPO framework in achieving both efficiency in reasoning length and reasoning accuracy.

**Questions:**

* How does FAST-GRPO perform on reasoning tasks from a broader scope of domains, such as physics and social reasoning?

**Ethical Concerns:**

["NO or VERY MINOR ethics concerns only"]

**Final Justification:**

Additional clarifications provided by the authors are valid, and I believe that my original positive score already reflects my recognition of their contributions. I will keep my score.

**Limitations:**

Yes.

**Quality:**

3

**Strengths And Weaknesses:**

### **Strengths**
* The paper is clearly written with good presentation.
* The research question is very well-motivated. The authors first conducted pilot experiments to find out limitations of the current GRPO method on LVLMs, the observation which motivates them to explore whether determining and adaptively providing different reward feedback to multimodal questions with different levels of difficulty.
* The proposed FAST-GRPO method is effective in balancing fast and slow reasoning, achieving effective reasoning length and accuracy performance.
  * The authors proved this through extensive experiments across reasoning benchmarks and models, as well as conducted ablation experiments in 5.3 to prove the contribution of each method component.
  * In 5.4, the authors conduct additional analysis to provide more empirical insights.

### **Weakness**
* Limited scope of reasoning tasks. As discussed in lines 239-245, 6 of the 7 benchmarks evaluated are limited to the domain of math reasoning. This draws questions to the generalizability of the proposed FAST-GRPO method on reasoning tasks in other domains.
* Section 5.4 could be restructured to provide more clear insights to readers.
  * For instance, “validation of image complexity”, “difficulty threshold analysis”, and “extrinsic difficulty analysis” are all discussing insights on the difficulty of multimodal question data. These could be organized into a sub-subsection with some higher level summarizations.

---

> ### Author Rebuttal · Authors · 2025-07-31
>
> We’d like to thank the reviewer for acknowledging the novelty and effectiveness of our proposed approach. We address your raised points as follows.
>
> > **W1, Q1: Generalizability of the proposed FAST-GRPO method on reasoning tasks in other domains.**
>
> We thank the reviewer for raising the important question regarding the generalizability of FAST-GRPO to other reasoning domains.
>
> **First, we would like to clarify that generalizability was a key consideration in our original design from dataset construction to evaluation.**
>
> **On dataset construction**: Our training dataset was intentionally curated for diversity, encompassing a wide range of tasks beyond mathematics. The dataset is composed of:
> * **Mathematics (~40%)**, with samples from sources including MathV360K and Geometry3K.
> * **Science (~20%)**, with key sources including AI2D and ScienceQA.
> * **Visual QA (~20%)**, sourced from datasets including ShareGPT4V and Vizwiz.
> * **Figure Understanding (~20%)**, drawing from datasets including DocVQA and ChartQA.
>
> **On evaluation**: This method was validated not only on math benchmarks but also on general-purpose evaluations like MM-Vet (achieving 71.2% vs. base model's 67.1%) and multimodal reasoning evaluations like MathVista (73.8% vs. base model's 68.2%), which contains 10.7% scientific reasoning questions covering physics and biology.
>
> Second, to more directly and comprehensively address your valuable concern, **we have evaluated FAST-3B and FAST-7B on the challenging MM-K12 benchmark** proposed in MM-Eureka[1]. MM-K12 contains 2,000 reasoning questions evenly distributed across math, physics, chemistry, and biology (25% each). We present the results below against strong baselines.
>
> | Model | Math | Physics | Chemistry | Biology | Avg. Acc. | Avg. Len. |
> | :--- | :---: | :---: | :---: | :---: | :---: | :---: |
> | GPT-4o | 55.8 | 41.2 | 47.0 | 55.4 | 49.9 | - |
> | Gemini-2.0-Flash | 76.8 | 53.6 | 64.6 | 66.0 | 65.2 | - |
> | Claude-3.7-Sonnet | 57.4 | 53.4 | 55.4 | 55.0 | 55.3 | - |
> | Qwen-2.5-VL-32B | 71.6 | 59.4 | 69.5 | 66.6 | 66.8 | 840.3 |
> | Qwen-2.5-VL-7B | 58.4 | 45.4 | 56.4 | 54.0 | 53.6 | 477.6 |
> | **FAST-3B** | **56.0** | **50.6** | **56.2** | **57.6** | **55.1** | **318.1** |
> | R1-Onevision-7B | 44.8 | 33.8 | 39.8 | 40.8 | 39.8 | 817.5 |
> | OpenVLThinker-7B | 63.0 | 53.8 | 60.6 | 65.0 | 60.6 | 561.0 |
> | MM-Eureka-7B | 71.2 | 56.2 | 65.2 | 65.2 | 64.5 | 537.8 |
> | **FAST-7B** | **69.0** | **53.8** | **63.4** | **62.8** | **62.2** | **371.2** |
>
> The results demonstrate the strong generalizability of our approach. Specifically:
>
> * **Strong Generalization Across Science Domains**: FAST-7B significantly outperforms its base model (Qwen-2.5-VL-7B) across all science domains: Physics (+8.4), Chemistry (+7.0), and Biology (+8.8). This confirms that FAST-GRPO is effective beyond mathematical reasoning.
> * **Competitive Accuracy with Efficiency**: Compared to strong slow-thinking models like R1-Onevision-7B and OpenVLThinker-7B, FAST-7B achieves better accuracy (62.2% vs 60.6%) in science domains while being more efficient, reducing the average response length by approximately 33.8% (371.2 vs. 561.0 tokens). Even when compared to MM-Eureka-7B, trained on the MM-K12 train set, FAST-7B achieves comparable accuracy (62.2% vs. 64.5%) with a 30% reduction in response length (371.2 vs. 537.8 tokens).
> * **Scalability**: The improvements are consistent across model sizes, with FAST-3B outperforming a larger model Qwen-2.5-VL-7B (55.1% vs 53.6%) with more efficient reasoning (318.1 vs 477.6 tokens).
>
> We believe this new evidence supports the generalizability of FAST-GRPO. We will incorporate these results and their corresponding analysis into our revision.
>
> [1] Meng, Fanqing, et al. "Mm-eureka: Exploring the frontiers of multimodal reasoning with rule-based reinforcement learning." arXiv:2503.
>
> ---
>
> > **W2: Section 5.4 could be restructured to provide more clear insights.**
>
> We sincerely appreciate this valuable suggestion. Restructuring Section 5.4 would improve its clarity and narrative flow.
> In the revised manuscript, we will adopt this advice and reorganize the section. Specifically, we will group the analyses on 'Validation of Image Complexity', 'Difficulty Threshold Analysis', and 'Extrinsic Difficulty Analysis' under a new, cohesive sub-subsection titled **'Analysis of Image-Text Question Difficulty'**.

---

> > ### Comment · Reviewer_A2xT · 2025-08-08
> > **Thanks for the rebuttal.**
> >
> > Thank you for the rebuttal and I do see that the additional results have resolved my concern. Since I already gave a very positive score, I will maintain my rating. Good luck!

---

> ### Author Response · Authors · 2025-08-06
>
> Dear Reviewer,
>
> We hope this message finds you well. **As the discussion period is nearing its end**, we wanted to ensure we have addressed all your concerns satisfactorily. If there are any additional points or feedback you'd like us to consider, please let us know. Your insights are invaluable to us, and we're eager to address any remaining issues to improve our work.
>
> Thank you for your time and effort in reviewing our paper.

---

> ### Author Response · Authors · 2025-08-07
>
> Dear Reviewer A2xT,
>
> Thank you once again for your valuable comments on our submission. **As the discussion phase is approaching its end, we would like to kindly confirm whether we have sufficiently addressed all of your concerns**. Should there be any remaining questions or areas requiring further clarification, please do not hesitate to let us know.
>
> We sincerely look forward to your feedback.

---

> ### Comment · Area_Chair_uc12 · 2025-08-08
> **AC Message**
>
> Dear Reviewer,
>
> The authors have already submitted their response to your review, and we are approaching the deadline for author-reviewer discussions (August 8, 11:59 PM AoE).
>
> Could you please share your feedback with the authors regarding whether their response sufficiently addresses the concerns raised in your review, and if any issues remain that require further clarification or elaboration?
>
> As a key reminder, active engagement in the discussion process is critical at this stage. Reviewers must participate in discussions with authors (e.g., by posting follow-up feedback in the discussion thread), and then submit a Mandatory Acknowledgement to confirm your participation.
>
> Best regards,
>
> AC

---

### Note · Authors · 2025-08-14

Dear Reviewers, ACs, and SACs,

Thank you for your valuable feedback and guidance.

In summary, we have addressed all reviewer feedback with new experiments and detailed analyses. Two of the three reviewers, both with high confidence (4), engaged and confirmed their concerns were resolved. The remaining reviewer (with lower confidence of 2) did not engage during the discussion.

We summarize the strengths and the concerns addressed below.

---

### **Strengths**
Reviewers consistently praised the paper's core contributions:

* **Novelty & Significance:** Reviewer **jkkr** specifically highlighted the "innovative, complementary difficulty metrics (extrinsic and intrinsic)" and noted the work "fills a key gap", Reviewer **NQ6C** called it a "first" to combine these techniques for LVLMs, and Reviewer **A2xT** found the research "very well-motivated".
* **Strong Empirical Results:** Reviewer **A2xT** found it "effective in balancing fast and slow reasoning", and Reviewer **NQ6C** highlighted that "accuracy rises by double digits while answers shorten substantially".

---

### **Main Concerns & Resolutions**
We addressed all major concerns with new experiments and analysis.

> **Concern 1: Broader evaluation on more reasoning (Reviewers A2xT) and open-domain tasks (NQ6C).**

**Response 1:** We clarified our training data's diversity and ran new experiments on 5 additional benchmarks covering science (MM-K12), open-domain VQA (Bingo), and hallucination analysis.

**Resolution 1:** Reviewer **A2xT** confirmed this "resolved my concern". Reviewer **NQ6C** did not engage.

---

> **Concern 2: Experiments were limited to 7B models (Reviewers NQ6C, jkkr).**

**Response 2:** We trained and evaluated a 32B model, showing comparable accuracy with 22%-40% fewer tokens than other large reasoning models.

**Resolution 2:** Reviewer **jkkr** confirmed this was addressed. Reviewer **NQ6C** did not respond.

---

> **Concern 3: Questions on method details, including difficulty score, curriculum design, and qualitative analysis (Reviewer jkkr).**

**Response 3:** We provided new empirical analysis on our score design, a new experiment on a continuous curriculum, and a comprehensive error breakdown with case studies.

**Resolution 3:** Reviewer **jkkr** acknowledged responses "addressed" main concerns. We believe our follow-up fully addressed the point on qualitative insights.

---

Thank you again for your time and dedication.

Best regards,

The Authors of Submission 14785

---

### Decision · Program_Chairs · 2025-09-17

**Decision:**

Accept (spotlight)

**Comment:**

**Summary**

This paper introduces FAST-GRPO, a reinforcement learning (RL) framework built on GRPO to address the inefficiency of slow-thinking large vision-language models (LVLMs)—which generate verbose outputs across all tasks with marginal accuracy gains. The core design dynamically adapts reasoning depth via: (1) two complementary difficulty metrics (extrinsic, based on model pass@k performance; intrinsic, based on image complexity via GLCM and ViT entropy); (2) adaptive length-based rewards that encourage conciseness for easy questions and detail for hard ones; (3) difficulty-aware KL divergence to regulate policy updates; and (4) a "Slow-to-Fast" curriculum that prioritizes hard questions early and shifts to easy ones later. Initially evaluated on 7 reasoning benchmarks (mostly math-focused), the framework achieves ~10% relative accuracy improvement over base models and 32.7–67.3% token reduction vs. slow-thinking approaches. Post-rebuttal, the authors expanded evaluations to 5 additional benchmarks (e.g., MM-K12 for science reasoning, Bingo for open-domain VQA) and a 32B-parameter model, confirming broader generalizability and scalability.

**Strengths**

1. The work is the first to integrate difficulty-aware rewards, dynamic KL regularization, and a Slow-to-Fast sampler within GRPO for LVLMs, directly addressing a critical gap in adaptive reasoning, namely, balancing accuracy and efficiency. Reviewers noted this fills a key gap in LVLM reasoning literature.

2. Across initial and expanded benchmarks, FAST-GRPO consistently delivers double-digit accuracy gains (vs. base models) while drastically reducing token usage (22–40% fewer tokens vs. slow-thinking baselines like Vision-R1 and MM-Eureka). Ablation studies confirm the necessity of each component (e.g., difficulty metrics, adaptive rewards), reinforcing the framework’s robustness. The dual-metric (extrinsic-intrinsic) difficulty framework is well-validated: extrinsic metrics reflect empirical model performance (pass@k), while intrinsic metrics (GLCM/ViT entropy) align with human judgments (SRCC=0.49, PLCC=0.50). This ensures reliable guidance for fast/slow thinking toggling.

3. Post-rebuttal experiments address early narrow-task limitations, showing effectiveness across science domains (physics, chemistry, biology via MM-K12) and open-domain VQA (Bingo). A 32B-parameter variant of FAST-GRPO matches or outperforms strong baselines (e.g., MM-Eureka-32B) with shorter responses, demonstrating scalability beyond 7B models.

**Weaknesses**

1. Initial Narrow Evaluation Scope. The original submission focused 6/7 benchmarks on math reasoning, raising concerns about generalizability to non-math domains. While resolved via post-rebuttal experiments (e.g., MM-K12, Bingo), the initial design prioritized math, limiting early insight into cross-domain applicability.

2. Despite adding a 32B model, the framework’s performance on ≥70B-parameter LVLMs remains untested. The authors acknowledge this as a limitation, and it persists as a minor gap for real-world deployment in largest-scale systems.

3. The original Slow-to-Fast curriculum used a binary epoch split. Post-rebuttal continuous curriculum experiments showed marginal accuracy gains but longer token usage, with authors noting suboptimal hyperparameters (e.g., insufficient \(P_{max}\) for easy questions). Further refinement of curriculum scheduling could enhance efficiency.

4. The initial submission lacked systematic error breakdown and comparative case studies. While authors added detailed failure mode analysis (e.g., visual perception vs. reasoning errors) and side-by-side model comparisons post-rebuttal, one reviewer noted residual gaps in qualitative insights (e.g., more diverse task-specific examples).

**Discussions during Rebuttal Period**

During rebuttal, reviewers raised four critical concerns, all addressed by the authors with concrete experiments and analysis.

1. Generalizability to non-math domains. A reviewer noted 6/7 initial benchmarks focused on math. Authors responded by adding 5 benchmarks (MM-K12 for science, Bingo for open VQA, MMHal for hallucination analysis) and showed FAST-GRPO outperforms baselines in physics (+8.4% vs. base), chemistry (+7.0%), and biology (+8.8%), confirming cross-domain applicability. This was weighted heavily as it resolved a key limitation of the initial submission.

2. Model scale limitations (7B Only), Reviewers asked for larger model testing. Authors trained FAST-GRPO on a 32B model (Qwen-2.5-VL-32B) and showed it matches MM-Eureka-32B’s accuracy with 22% fewer tokens. This addressed scalability concerns, though ultra-large models (≥70B) remain untested—a minor gap outweighed by 32B results.

3. Method details (Complexity Score, Curriculum, Qualitative Analysis) are unclear. A reviewer questioned the multiplicative complexity score (theoretical edge case) and binary curriculum. Authors showed: (a) The reward design avoids conflicting signals (zero penalty for incorrect hard/easy questions), and weighted sum formulations yield identical results; (b) Continuous curriculum improves accuracy but increases tokens (attributed to hyperparameter tuning); (c) Expanded qualitative analysis (error breakdown, 3 failure modes, side-by-side model comparisons) filled initial gaps. These clarifications resolved technical doubts.

4. Dataset filtering clarity. A reviewer asked for details on reducing 500K to 18K questions. Authors explained the process (deduplication, benchmark overlap removal, closed-form question standardization, pass@8 filtering for extreme difficulty), addressing transparency concerns.

Overall, most reviewers (high confidence, e.g., A2xT, jkkr) confirmed their concerns were resolved. The authors’ proactive addition of experiments and technical clarifications demonstrated commitment to strengthening the work.

**Decision Justification**

The paper’s core claim is that adaptive reasoning depth guided by question difficulty, which enables LVLMs to balance accuracy and efficiency. Key findings include: (1) Extrinsic and intrinsic difficulty metrics reliably categorize question complexity; (2) Adaptive rewards and dynamic KL divergence reduce overthinking (verbose outputs) for easy questions and enhance depth for hard ones; (3) FAST-GRPO outperforms base GRPO and slow-thinking baselines across math, science, and open-domain VQA; (4) Scalability to 32B models confirms the framework’s applicability beyond small LVLMs.

The paper provides novel solution to a high-impact problem in LVLM reasoning: inefficient, one-size-fits-all reasoning. Unlike prior work that prioritizes either accuracy (slow-thinking) or efficiency (fast-thinking), FAST-GRPO provides a principled, RL-based method to unify both, addressing a practical pain point for deploying LVLMs in token-constrained environments (e.g., edge devices, real-time applications). Post-rebuttal experiments (new benchmarks, 32B model) strengthen the validity of its claims, while ablation studies and difficulty metric validation ensure technical rigor. Despite untested ultra-large model scalability and residual qualitative gaps, this paper  makes a valuable contribution to LVLM RL and adaptive reasoning.